# Hypothalamic CNTF volume transmission shapes cortical noradrenergic excitability upon acute stress

Alán Alpár[1,2,†,‡,*], Péter Zahola[1,2,†], János Hanics[1,2], Zsófia Hevesi[1], Solomiia Korchynska[3], Marco Benevento[3], Christian Pifl[3], Gergely Zachar[2], Jessica Perugini[4], Ilenia Severi[4], Patrick Leitgeb[3], Joanne Bakker[5], Andras G Miklosi[3], Evgenii Tretiakov[6], Erik Keimpema[3], Gloria Arque[3], Ramon O Tasan[7], Günther Sperk[7], Katarzyna Malenczyk[3], Zoltán Máté[8], Ferenc Erdélyi[8], Gábor Szabó[8], Gert Lubec[9], Miklós Palkovits[2,10], Antonio Giordano[4], Tomas GM Hökfelt[5], Roman A Romanov[3,6,‡], Tamas L Horvath[11,12,‡] & Tibor Harkany[3,4,‡,**] [iD]

## Abstract

Stress-induced cortical alertness is maintained by a heightened excitability of noradrenergic neurons innervating, notably, the prefrontal cortex. However, neither the signaling axis linking hypothalamic activation to delayed and lasting noradrenergic excitability nor the molecular cascade gating noradrenaline synthesis is defined. Here, we show that hypothalamic corticotropin-releasing hormone-releasing neurons innervate ependymal cells of the 3rd ventricle to induce ciliary neurotrophic factor (CNTF) release for transport through the brain's aqueductal system. CNTF binding to its cognate receptors on norepinephrinergic neurons in the locus coeruleus then initiates sequential phosphorylation of extracellular signal-regulated kinase 1 and tyrosine hydroxylase with the $Ca^{2+}$-sensor secretagogin ensuring activity dependence in both rodent and human brains. Both CNTF and secretagogin ablation occlude stress-induced cortical norepinephrine synthesis, ensuring neuronal excitation and behavioral stereotypes. Cumulatively, we identify a multimodal pathway that is rate-limited by CNTF volume transmission and poised to directly convert hypothalamic activation into long-lasting cortical excitability following acute stress.

**Keywords** $Ca^{2+}$-sensor protein; hypothalamus–pituitary–adrenal axis; neurotrophin; post-traumatic stress disorder; prefrontal cortex
**Subject Categories** Neuroscience
The EMBO Journal (2018) 37: e100087

See also: **D Pozzi & M Matteoli** (November 2018)

## Introduction

Ensuring species' survival in perilous environments is a primary evolutionary demand. Therefore, central stress pathways, efficiently linking the brain and periphery, have evolved to form the hypothalamus–pituitary–adrenal axis (HPA) (Selye & Fortier, 1949; Bale & Vale, 2004; McEwen, 2008). By principle, corticotropin-releasing

1 SE NAP Research Group of Experimental Neuroanatomy and Developmental Biology, Semmelweis University, Budapest, Hungary
2 Department of Anatomy, Histology, and Embryology, Semmelweis University, Budapest, Hungary
3 Department of Molecular Neurosciences, Center for Brain Research, Medical University of Vienna, Vienna, Austria
4 Section of Neuroscience and Cell Biology, Department of Experimental and Clinical Medicine, Marche Polytechnic University, Ancona, Italy
5 Department of Neuroscience, Karolinska Institutet, Stockholm, Sweden
6 Immanuel Kant Baltic Federal University, Kaliningrad, Russia
7 Department of Pharmacology, Medical University Innsbruck, Innsbruck, Austria
8 Institute of Experimental Medicine, Hungarian Academy of Sciences, Budapest, Hungary
9 Paracelsus Medical University, Salzburg, Austria
10 Human Brain Tissue Bank and Laboratory, Semmelweis University, Budapest, Hungary
11 Program in Integrative Cell Signaling and Neurobiology of Metabolism, Departments of Comparative Medicine and Neuroscience, Kavli Institute for Neuroscience, Yale University School of Medicine, New Haven, CT, USA
12 Department of Anatomy and Histology, University of Veterinary Medicine, Budapest, Hungary
*Corresponding author. Tel: +36 1 2156 920/53609; E-mail: alpar.alan@med.semmelweis-univ.hu
**Corresponding author (Lead contact). Tel: +43 1 40160 34050; E-mail: tibor.harkany@meduniwien.ac.at
†These authors contributed equally to this work as first authors
‡These authors contributed equally to this work as senior authors
[The copyright line of this article was changed on 26 November 2018 after original online publication.]

hormone (CRH)-containing($^+$) neurons of the paraventricular hypothalamic nucleus (PVN) gate output from the central nervous system (Swanson & Sawchenko, 1980) with CRH (Swanson *et al*, 1983, 1986), through pituitary amplification steps, triggering corticosteroid release from the adrenal glands for immediate metabolic mobilization (Rivier & Vale, 1983; Kovacs & Sawchenko, 1996). Nevertheless, stress, whether due to, e.g., predation or competition for reproduction, is unlikely a singular event. Therefore, secondary response pathways to enable an individual's prolonged vigilance might have evolved to confer added evolutionary benefit when responding to recurrent challenges. Seminal studies (Schulkin *et al*, 1994; McEwen & Sapolsky, 1995; Popoli *et al*, 2011) support this notion by documenting that corticosteroids released from the adrenals do not only produce feedback inhibition of hypothalamic and pituitary hormone secretion (Akana *et al*, 1992) but directly regulate limbic and reward circuits (Sapolsky, 2003) to gate coping and flexibility (e.g., "flight or fight" behaviors; Eriksen *et al*, 1999; McEwen *et al*, 2012), motivation (McEwen, 2005), memory (Roozendaal *et al*, 2009), and fear extinction (Korte, 2001; McEwen, 2005). The prefrontal cortex (PFC) has emerged as a central site to orchestrate coordinated responses to acute stress (McEwen, 2007) with glucocorticoid and mineralocorticoid receptors modulating its ability to integrate upstream emotional, sensory, cognitive, and spatial inputs (Patel *et al*, 2008; Gadek-Michalska *et al*, 2013; Caudal *et al*, 2014).

In general terms, corticosteroids can change neuronal excitability through the cell-type-specific engagement of mineralocorticoid and glucocorticoid receptors with the former increasing and the latter suppressing neuronal activity (Joels & de Kloet, 1992). However, stress-induced alertness (defined as heightened cortical network excitability for prolonged periods) might benefit from a single neural trigger, such as CRH$^+$ neuroendocrine cells, for the tight temporal coupling and scaling of cortical excitability for conscious execution of stereotyped behaviors associated with vigilance and HPA-induced peripheral energy mobilization. A neural link between CRH$^+$ parvocellular cells and norepinephrinergic (NE) neurons of the locus coeruleus (LC) is of particular appeal because NE activity increases with the severity and duration of stress (Aston-Jones *et al*, 1996; Chowdhury *et al*, 2000) and NE afferents of the PFC are poised to facilitate adaptive behaviors (Uematsu *et al*, 2017). Recently, CRH$^+$ innervation of NE neurons has been described (Zhang *et al*, 2017), including at the ultrastructural level (Van Bockstaele *et al*, 1996). However, a monosynaptic circuit operating through excitatory CRH [i.e., CRH acting at $G_s$ protein-coupled *Crhr1* and/or *Crhr2* receptors (De Souza, 1995)] seems insufficient to functionally convert short-lived surges of excitability into long-lasting NE sensitization for cortical stress adaptation, particularly since neuropeptide release likely commences only upon intense burst firing (Overton & Clark, 1997).

Here, we unmask an efficient mechanism coordinated by glutamate release from CRH neurons onto ependymal cells that line the wall of the 3$^{rd}$ ventricle to trigger long-range volume transmission by ciliary neurotrophic factor (CNTF) in the brain aqueductal system. Once reaching the LC, CNTF heightens NE output (Fig 1A), as opposed to fast synaptic coupling known to evoke anxiety acutely (Zhang *et al*, 2017). We show the maintenance of NE excitability through CNTF-induced sequential recruitment of secretagogin (*Scgn*) and extracellular signal-regulated kinase 1 (*Erk1*) to increase tyrosine hydroxylase (TH) activity by phosphorylation for cortical NE production. Despite extensive NE innervation of the entire cortical mantle (Fuxe *et al*, 1968; Moore & Bloom, 1979; Aston-Jones,

1995), this mechanism centers on the PFC where it is poised to efficiently reset network excitability (Fig 1A; McCormick *et al*, 1991). Thus, the combination of genetic manipulation of *Cntf* and *Scgn* with opto-/chemogenetics and biochemistry not only uncovers previously undescribed molecular determinants gating stress-induced behavioral phenotypes but also offers targets for stress resilience.

## Results

### Ependymal cells are an intrahypothalamic target of CRH neurons

Paraventricular CRH neurons of the hypothalamus release CRH into the median eminence to control the HPA stress axis by facilitating adrenocorticotropic hormone (ACTH) release from the anterior pituitary (Bale & Vale, 2004). However, whether paraventricular CRH neurons project to other targets within the hypothalamus, as proposed for other types of parvocellular neurons (Ter Horst & Luiten, 1987; Dai *et al*, 1998), remains undefined. First, we addressed alternative synaptic sites for CRH neurons by microinjecting adeno-associated virus (AAV8) particles encapsulating an mCherry reporter into adult *Crh-Ires*-Cre mice that are commonly used to test stress-related behaviors (Fuzesi *et al*, 2016). By postoperative days 5–7, mCherry-labeled processes emanating from CRH neurons that reside in the PVN coursed toward the 3$^{rd}$ ventricle (Fig 1B) with mCherry$^+$ bouton-like varicosities lining the outermost ependymal layer of the 3$^{rd}$ ventricle. Next, we tested whether ependymal cells could directly respond to synaptic signals of CRH$^+$ neurons by using single-cell RNA-seq to survey their CRH, glutamate, and GABA receptor contents (Romanov *et al*, 2017b). Ependymal cells, clustered by their expression of *Enkur* and *Foxj1* protogenes (Romanov *et al*, 2017b), predominantly expressed mRNA transcripts for glutamate and select GABA$_A$ receptor subunits (Fig 1C) with unexpectedly sparse mRNA content for *Crhr1* and *Crhr2* receptors. These data suggest that ependymal cells could respond to glutamate (co-)released from "stress-on" CRH$^+$ neuroendocrine cells (Romanov *et al*, 2015, 2017b).

We have developed *Crh-Ires*-Cre::*egfp* mice to demonstrate that EGFP$^+$ nerve endings contained vesicular glutamate transporter 2 (VGLUT2; Fig EV1A and A1) and less so VGLUT1 (Fig EV1A) along the 3$^{rd}$ ventricle wall, suggesting the likelihood of glutamate release from CRH$^+$ terminals. We then confirmed that VGLUT2$^+$ nerve endings apposed ependymal cells that expressed GRIA1 (Fig 1C1), the α-amino-3-hydroxy-5-methyl-4-isoxazole propionate (AMPA) receptor subunit most abundantly expressed by ependymal cells at the mRNA level (Fig 1C). Notably, our three-dimensional tissue reconstructions revealed that only a subset of ependymal cells received VGLUT2$^+$ innervation (Fig 1C1), which could preclude their widespread and synchronous synaptic activation. However, ultrastructural analysis demonstrated that ependymal cells in the dorsolateral segment of the 3$^{rd}$ ventricle wall are connected by gap junctions (Fig 1D1) with their plasmalemma often convoluted (Fig EV1B) to increase surface contact (Vanslembrouck *et al*, 2018). These data were substantiated by dye transfer from biocytin-loaded ependymal cells to their closest neighbors (Fig 1D2). Thus, gap junctions are the structural basis to convert the synaptic activation of "first-responder" ependymal cells to synchronous cell-state changes in a larger ependymal network.

We used patch-clamp electrophysiology *ex vivo* to monitor whether ependymal cells receive synaptic inputs. Firstly, ependymal

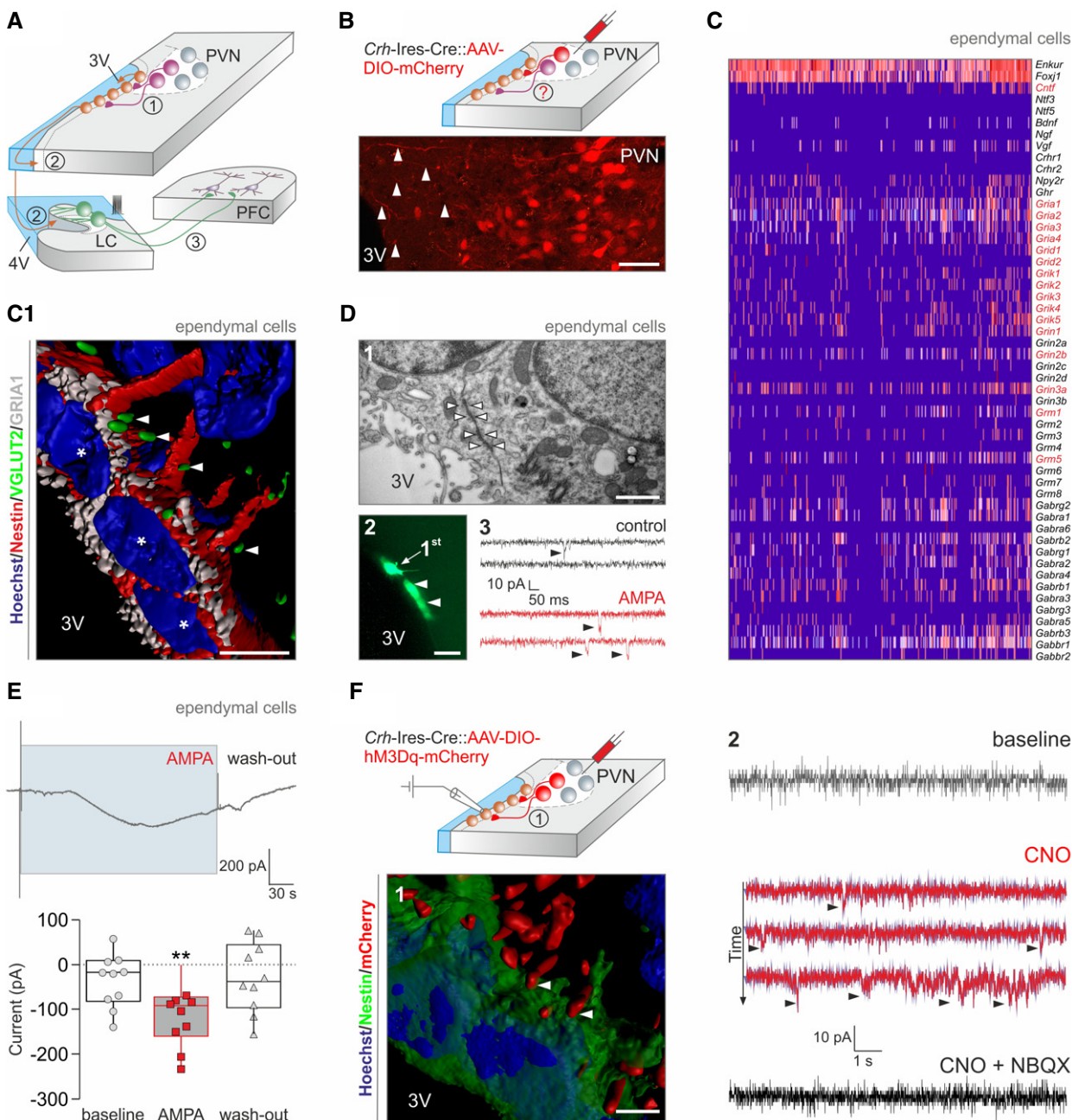

**Figure 1.  Hypothalamic corticotropin-releasing hormone (CRH)-releasing neurons innervate ependymal cells lining the 3ʳᵈ ventricle.**

A   Cartoon depicting a multimodal signaling axis including a direct pathway between the paraventricular hypothalamic nucleus (PVN) and ventricular ependyma (1), volume transmission to the locus coeruleus (LC; 2) with norepinephrinergic projections to the prefrontal cortex (PFC; 3).

B   Microinjection of AAV-DIO-mCherry virus particles into the PVN of *Crh*-Ires-Cre mice reveals mCherry-containing processes oriented toward the 3ʳᵈ ventricle (3V; *arrowheads*). Scale bar = 60 μm.

C   Single-cell RNA-seq reveals infrequent expression of *Crhr1, Crhr2*, and *Crhbp*, as opposed to glutamate and GABA receptor subunits (in *red*), by ependymal cells. Ependymal cells were classified by *Enkur* and *Foxj1* expression (Romanov *et al*, 2017b), and also contained *Cntf* mRNAs. (C1) Reconstruction of GRIA1⁺ ependymal cells receiving VGLUT2⁺ synapses (*arrowheads*). Asterisks denote nuclei. Scale bar = 12 μm.

D   (1) Electron micrograph showing gap junction coupling (*arrowheads*) between ependymal cells. Scale bar = 250 nm. (2) Dye transfer among ependymal cells. "1ˢᵗ" indicates the cell probed directly. Arrowheads indicate secondary labeling in adjacent cells. Scale bar = 15 μm. (3) Postsynaptic currents (*arrowheads*) recorded in ependymal cells in control and upon exposure to AMPA (10 μM).

E   *Upper panel*: Tonic inward current produced by bath-applied AMPA (10 μM). *Lower panel*: Quantitative data from ependymal cells from *n* > 3 mice. Data in box plots represent medians and 10ᵗʰ, 25ᵗʰ, 75ᵗʰ, and 90ᵗʰ percentiles. **P < 0.01 vs. baseline and wash-out (ANOVA).

F   Activating DREADD (hM3Dq) was microinjected into the PVN of *Crh*-Ires-Cre mice 14–17 days prior to *ex vivo* recordings. (1) Reconstruction of mCherry-labeled terminals (*arrowheads*) in apposition to nestin⁺ ependyma. Scale bar = 7 μm. (2) DREADD activation by CNO in CRH terminals innervating ependymal cells induces inward currents (*arrowheads*), which are sensitive to NBQX, an AMPA receptor antagonist (20 μM).

cells (for basic membrane properties, see Fig EV1C–C3) produced spontaneous postsynaptic currents, which increased in frequency when bath-applying AMPA (10 μM; Figs 1D3 and EV1D–D3). Secondly, they invariably responded to AMPA superfusion by generating long-lasting inward currents when held at −70 mV (Fig 1E). We then addressed whether glutamatergic innervation of ependymal cells originates from CRH neurons by microinjecting adeno-associated virus (AAV) particles carrying Cre-dependent activating DREADD (hM3Dq) in tandem with an mCherry reporter (Alexander *et al*, 2009) into the PVN (Fig 1F). Histochemical localization of mCherry recapitulated the distribution of VGLUT2$^+$ synaptic puncta along ventricular ependyma (Fig 1F1), supporting the existence of a direct projection from parvocellular CRH neurons. Thereafter, we applied the DREADD agonist clozapine N-oxide (CNO, 10 μM) to acute brain slices to show the emergence of inward currents in ependymal cells, which were completely abolished by superfusion of 2,3-dihydroxy-6-nitro-7-sulfamoyl-benzo[f]chinoxalin-2,3-dion (NBQX, 20 μM), an AMPA receptor antagonist (Fig 1F2). Overall, these data suggest that ependymal cells along the anterior-dorsal segment of the 3$^{rd}$ ventricle are intrahypothalamic targets for CRH neurons and are tonically excited by glutamate.

## Glutamatergic neurotransmission facilitates CNTF release into the cerebrospinal fluid upon acute stress

The existence of a monosynaptic pathway originating from CRH$^+$ neurons and innervating ventricular ependyma points to the stress-induced release of bioactive substances into the cerebrospinal fluid. *Crh-Ires*-Cre::*egfp* mice were informative to reveal the genuine extent of EGFP$^+$ innervation within the proximity (< 15 μm) of the wall of the 3$^{rd}$ ventricle through lifetime synapse labeling (Fig 2A). In turn, quantitative histochemistry for CRH showed that acute formalin stress significantly increases the density of CRH$^+$ boutons targeting the wall of the 3$^{rd}$ ventricle (in rats: 6.93 ± 0.67 in control vs. 13.41 ± 0.93 20 min after stress, *P* < 0.05; Fig 2A1; Appendix Fig S1A). Subsequently, we used *cfos*-Cre$^{ERT2}$::*ROSA26-stop-ZsGreen1$^{f/f}$* mice in an activity "TRAP" approach (Guenthner *et al*, 2013; Fig 2B), as well as histochemical detection of c-Fos itself (Fig 2C; Appendix Fig S1B) to show that formalin stress significantly increases the density of both *ZsGreen1*$^+$ (at 48 h; Fig 2B1) and c-Fos$^+$ (at 2 h; Fig 2C) ependymal cells that co-express glial fibrillary acidic protein$^+$ (Fig 2B1). The localization of stress-activated ependymal cells to the cranial domain of the 3$^{rd}$ ventricle (*P* < 0.05; Appendix Fig S1C) resembles that of "light cells", which carry stereocilia and supposedly secrete bioactive substrates into the ventricular space (Mitro, 1981).

Ependymal cells also expressed *Cntf* mRNA (Fig 1C), a neurotrophin implicated in neurogenesis and repair (Kazim & Iqbal, 2016). We validated these data by anti-ciliary neurotrophic factor (CNTF) histochemistry (Severi *et al*, 2012; Fig 2D and D1), by RT–PCR in micropunches from the ventricular wall (Appendix Fig S1D), and by comparing cultured neurons and astroglia (Appendix Fig S1D1). High-resolution imaging showed EGFP$^+$ terminals in *Crh-Ires*-Cre::*egfp* mice in close apposition to CNTF$^+$ ependymal cells (Fig 2D2). Because acute stress increases CRH$^+$ synaptic input on ependymal cells (Fig 2A1), we measured whether this translates into CNTF being liberated into the cerebrospinal fluid (Appendix Fig S1E). Indeed, an ~threefold increase in CNTF levels was observed 20 min after acute stress in liquor collected from the 4$^{th}$ ventricle

(59.04 ± 26.41 pg/ml in control vs. 220.27 ± 98.51 in stress, *P* < 0.05; Fig 2E). Acute stress invariably evokes characteristic defensive behaviors, with an initial period of immobility ("freezing") followed by hypolocomotion (Morilak *et al*, 2005; de Andrade *et al*, 2012; Niermann *et al*, 2017) used as key indicators. If CNTF volume transmission mediates acute stress responses, then CNTF infusion into the cerebrospinal fluid under ambient conditions ought to mimic stress-induced behaviors. Indeed, intracerebroventricular (icv) CNTF infusion (4 μl, 6 ng/μl) in freely moving rats resulted in significant hypolocomotion (*P* < 0.05; Fig 2F). These data cumulatively suggest that stress-activated CRH neurons initiate phasic CNTF release into the aqueductal system for action by volume transmission in adult brain.

## CNTF activates tyrosine hydroxylase in norepinephrine neurons that project to the prefrontal cortex

CNTF released into the aqueductal system might act at sites distant from its place of production. We inferred that any neuronal contingent responding to CNTF stimuli must directly be exposed to the cerebrospinal fluid, likely through dendrites targeting the ependymal surface, and express CNTF tyrosine kinase receptors (CNTFRs; Ip *et al*, 1993). The open-source Allen brain atlas highlights CNTFR mRNA in the locus coeruleus (LC; Fig 3A). By using dual-label histochemistry, we demonstrate that tyrosine hydroxylase (TH)$^+$ processes targeting the overlaying ependymal surface of the 4$^{th}$ ventricle (Appendix Fig S2A) contain CNTFRs (Fig 3B and B1). We further integrated the LC into the neural circuitry responding to acute stress by showing stress-induced c-Fos immunoreactivity in this brain area (Fig 3C).

TH is the rate-limiting enzyme of NE production with its activity regulated by phosphorylation at Ser$^{31}$ (but not Ser$^{40}$; Zigmond *et al*, 1989; Dickson & Briggs, 2013; Tekin *et al*, 2014). Particularly, CNTF, like physical stressors (Ong *et al*, 2014), stimulates TH activity through Ser$^{31}$ phosphorylation (Shi *et al*, 2012b). Here, we find that recombinant CNTF induces significant Ser$^{31}$ phosphorylation of TH, along with prototypic Erk1 phosphorylation (Haycock *et al*, 1992) in cranial pons explants containing the LC (45 and 58% increase in the phosphorylated forms of TH and Erk, respectively, *P* < 0.05; Fig 3D–D2; Appendix Fig S2B). When applying subcutaneous formalin stress, Ser$^{31}$ phosphorylation of TH was transient (~twofold at 20 min, *P* < 0.05) and eluded its Ser$^{40}$ residue, which seems unrelated to augmenting NE synthesis upon pain-induced stress (Ong *et al*, 2014; Appendix Fig S2C and C1). To substantiate that ventricular ependyma is the source of CNTF to induce TH activity in LC, we performed site-directed *Cntf* silencing through icv siRNA infusion (Fig 3E). We then harvested LC tissues from animals that had been subjected to acute formalin stress 72 h after siRNA application and find the complete lack of Ser$^{31}$ phosphorylation of TH (Fig 3E1). In sum, we suggest that liquor-borne CNTF modulates NE output from the LC upon acute stress particularly since neither neurons nor glia prepared from cranial pons can produce CNTF and act as a local (and alternative) source (Appendix Fig S1D1).

NE neurons innervate the entire cortical mantle with particularly dense efferent projections to the PFC (Porrino & Goldman-Rakic, 1982; Nakane *et al*, 1994). PFC-projecting NE neurons were recently sub-classified functionally, being indispensable for behavioral flexibility (Uematsu *et al*, 2017). Considering the hypolocomotion

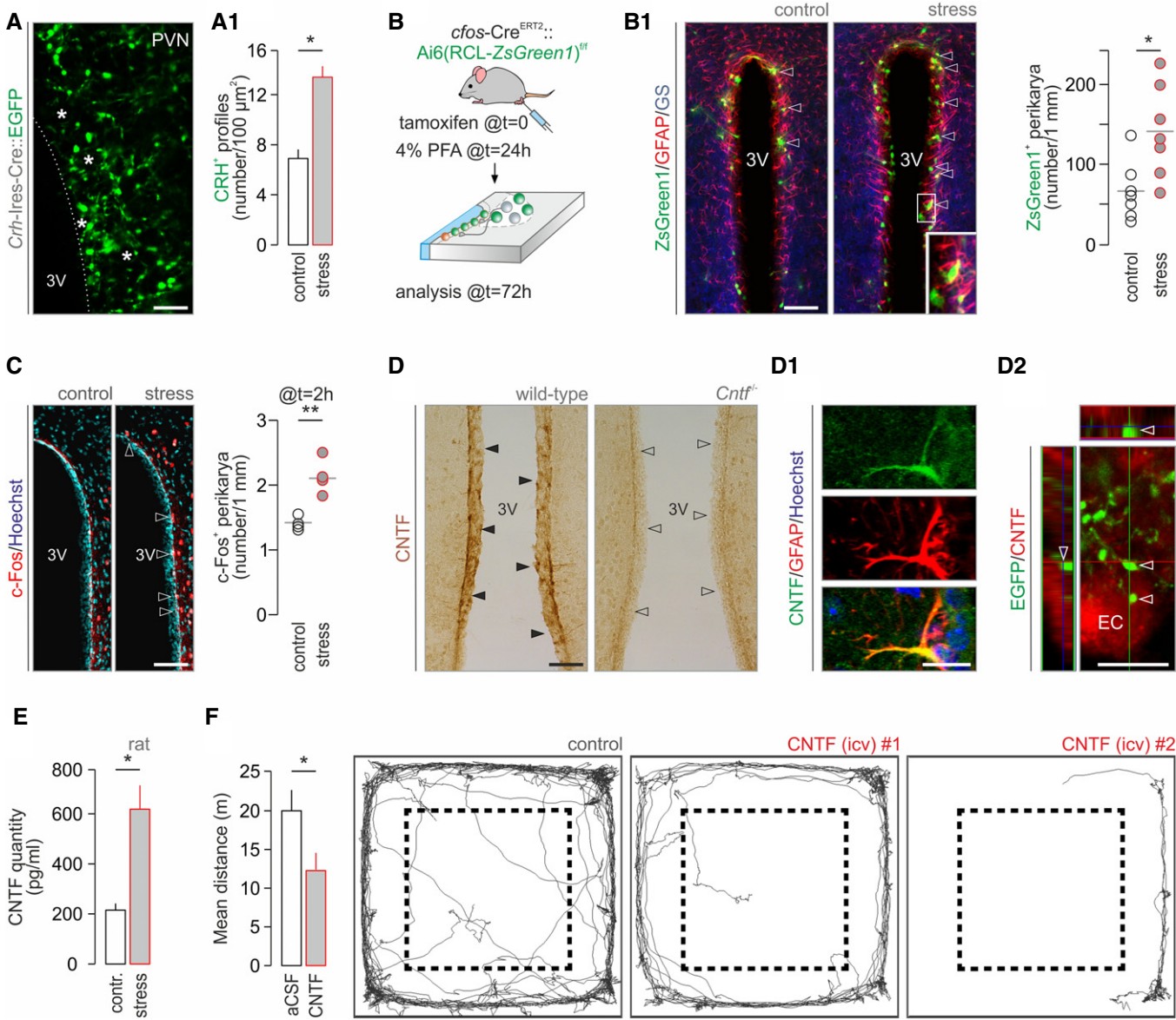

**Figure 2. Acute stress induces ciliary neurotrophic factor release from ependymal cells into the cerebrospinal fluid.**

A   CRH terminals around ependymal cells (asterisks) of the 3rd ventricle marked by lifetime GFP labeling in *Crh-Ires-Cre::egfp* mice ($n > 3$ mice/group). Scale bar = 10 μm. (A1) The density of CRH+ boutons is increased upon acute stress. *$P < 0.05$.

B   Experimental design of activity "TRAP" experiments using coincidence detection in *cfos-Cre*ERT2 mice. (B1) *Left*: Representative images of *ZsGreen1*+ ependymal cells in the dorsolateral domain of the 3rd ventricle wall, closest to the PVN, after acute stress (*arrowheads*). Inset identifies GFAP+/glutamine synthetase (GS)+ ependyma. Scale bar = 150 μm. *Right*: Quantitative analysis collected within a 15-μm-broad band around the 3rd ventricle. *$P < 0.05$, $n = 7$ mice/group.

C   Quantitative histochemistry for c-Fos 2 h after formalin stress. *Left*: Open arrowheads point to the increased density of c-Fos+ ependymal lining the 3rd ventricle. $n = 4$ mice/group. Scale bar = 150 μm. *Right*: Quantitative data on ependymal cells directly exposed at the ventricular surface. **$P < 0.01$.

D   Ependymal cells express CNTF in wild-type mice (*solid arrowheads*). *Cntf*−/− mice were used to validate our histochemical results with their ependyma devoid of CNTF-like immunoreactivity (*open arrowheads*). (D1) CNTF+ ependyma co-express GFAP. Hoechst 33,342 was used as nuclear counterstain. (D2) EGFP+ nerve endings of EGFP-expressing CRH neurons appose ependymal cells (EC; *arrowheads*) lining the 3rd ventricle. Orthogonal projection. Scale bars = 120 μm (D), 5 μm (D1), and 8 μm (D2).

E   CNTF concentration in the liquor filling the 4th ventricle is ~threefold increased 20 min after acute noxious stress. $n = 4$ rats/group were used. *$P < 0.05$.

F   CNTF infusion (4 μl, 6 ng/μl) in the 3rd ventricle induces hypolocomotion. Dashed lines indicate the center–periphery boundary used. *$P < 0.05$, $n = 7$/group. Representative traces of ambulation over a period of 10 min are shown.

Data information: Data are expressed as means ± s.e.m. and were analyzed by either Student's *t*-test (A1, B1, and C) or Mann–Whitney *U*-test (F).

evoked by CNTF infusion, we posit that those NE neurons that innervate the PFC might be particularly endowed with CNTFRs. We have tested this hypothesis by microinjecting biotinylated dextran amine, serving as retrograde tracer, into the medial PFC (mPFC) and quantifying CNTFR density on NE neurons 7 days later (Fig 3F and F1). Indeed, we find an ~threefold enrichment in CNTFRs in NE

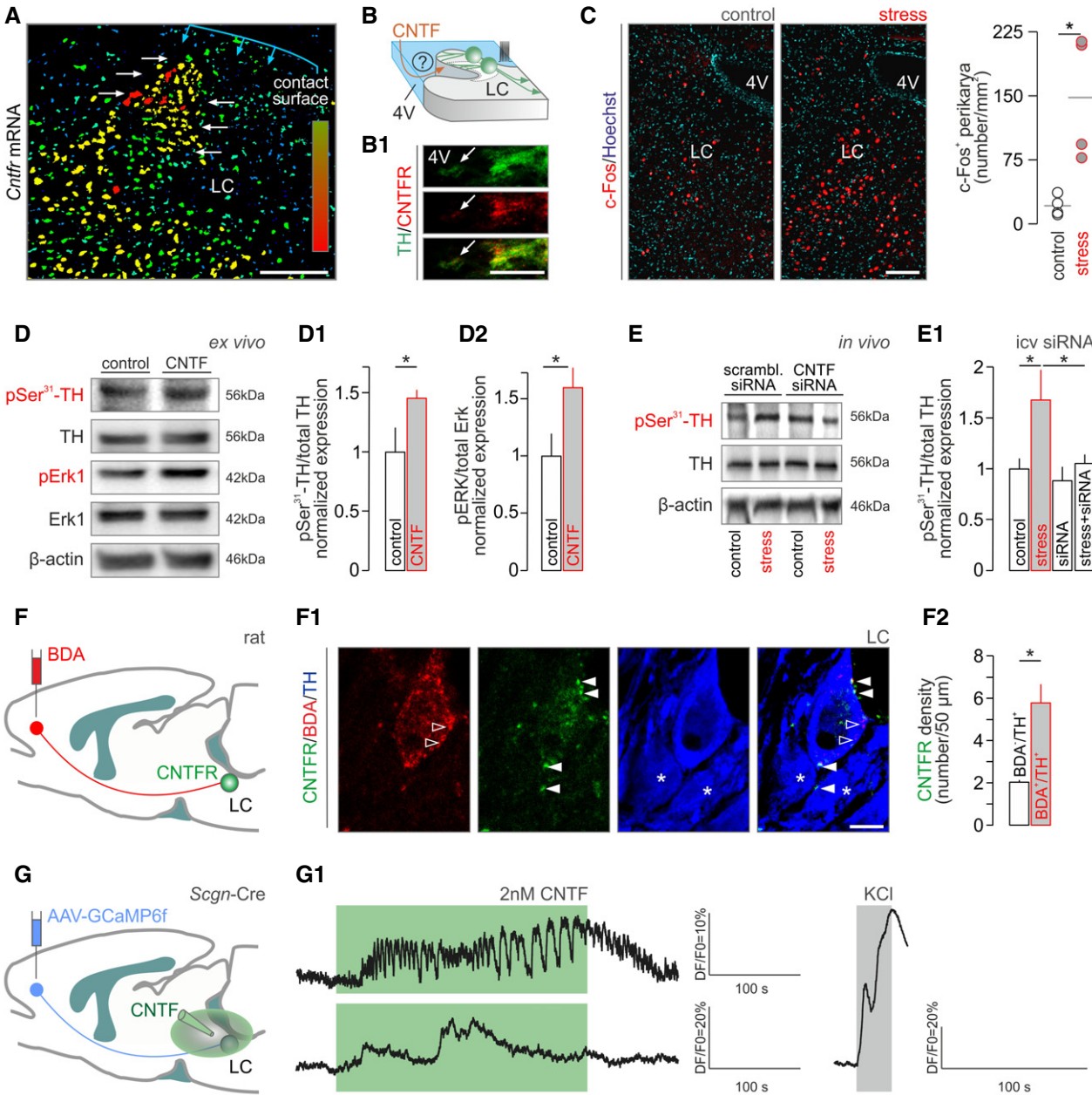

**Figure 3.  CNTF activates tyrosine hydroxylase by phosphorylation in mouse and human locus coeruleus.**

A   *Cntfr* mRNA expression in mouse locus coeruleus (LC, *white arrows*). Increasing mRNA content is indicated by a color gradient ranging from dark blue (low) to red (high). The ependymal surface as contact point with the 4th ventricle is shown in blue. Scale bar = 100 μm.

B   Hypothetical orientation of NE neurons with their dendrites emanating toward the 4th ventricle. (B1) CNTF receptor (CNTFR) localization in tyrosine hydroxylase (TH)+ processes extending toward the ventricular wall (arrow marks *ventricular surface*). Scale bar = 10 μm.

C   c-Fos activation in LC 2 h after formalin stress. *Left*: Increased density of c-Fos+ cells in the LC. Scale bar = 150 μm. *Right*: Quantitative data. *$P < 0.05$, $n = 4$/group.

D   Representative Western blots of *ex vivo* LC explants acutely exposed to recombinant CNTF (for 20 min) show increased TH (D1) and Erk1 (D2) phosphorylation ($n \geq 3$ samples/group). *$P < 0.05$.

E   Intracerebroventricular infusion of *Cntf* siRNA blunts stress-induced TH phosphorylation in the LC. (E1) Acute stress was triggered by injecting 4% PFA into a left hind paw unilaterally. Sampling followed 20 min later. *$P < 0.05$, $n = 8$/group.

F   Schematic diagram of the BDA-based retrograde labeling of LC neurons projecting to the mPFC. (F1) Multiple-labeling immunofluorescence detection of CNTFRs in a BDA+/TH+ neuron. *Solid arrowheads* point to perisomatic CNTFRs, and *open arrowheads* label BDA accumulation, while *asterisks* identify BDA−/TH+ neurons. Scale bar = 7 μm. (F2) Quantitative analysis of CNTFR density on the plasmalemmal surface of BDA− vs. BDA+ NE neurons in $n = 4$ rats. *$P < 0.05$.

G   Graphical rendering of the experimental outline to test CNTF effects on NE neurons projecting to the mPFC. GCaMP6f was microinjected into the mPFC *in vivo* with exogenous CNTF (2 nM) applied to brain slices containing the LC 14 days later. (G1) Representative traces of GCaMP6f fluorescence changes in two NE neurons in response to bath-applied CNTF (green shading). KCl (55 mM) was used as positive control.

Data information: Data are expressed as means ± s.e.m. and were statistically evaluated using Student's *t*-test (C, D1, D2, E1, F2).

neurons that provide efferentation to the mPFC (Fig 3F2). Finally, we tested whether mPFC-targeting NE neurons respond to CNTF by injecting AAVs encoding the $Ca^{2+}$ indicator GCaMP6 in a Cre-dependent manner into the PFC (Fig 3G). For this, we have developed secretagogin (*Scgn*)-Cre mice (Appendix Fig S2D and D1) because secretagogin, an EF-hand tetra-$Ca^{2+}$-sensor protein (Wagner *et al*, 2000; Alpár *et al*, 2012; Romanov *et al*, 2015), is abundantly expressed in NE neurons residing in LC (Mulder *et al*, 2009). Fourteen days after viral GCaMP6 delivery, we used $Ca^{2+}$ imaging to monitor whether CNTF superfusion affects NE neurons *ex vivo*. As Fig 3G1 shows, CNTF induced $Ca^{2+}$ influx in NE neurons projecting to the mPFC. Thus, we suggest that liquor-borne CNTF can modulate NE output in the mPFC upon acute stress.

## Secretagogin regulates tyrosine hydroxylase phosphorylation in NE neurons

We next interrogated whether CNTFR-dependent TH phosphorylation may be mediated by a hierarchical Erk1-dependent signaling cascade (Haycock *et al*, 1992). We inferred that the precise timing and duration of any such activation step is likely reliant on coincidence detectors, such as $Ca^{2+}$-sensor proteins (Romanov *et al*, 2015; Fig 4A). As such, NE neurons in both rodents (Figs 4B–B2 and EV2A–D) and humans (Fig 5A and A1) meet these criteria since they express secretagogin (Figs 4B1 and 5A1). Secretagogin is neuron-specific in mammalian brain (Mulder *et al*, 2009) and also coordinates acute stress-induced fast CRH release from parvocellular neurons at the median eminence (Romanov *et al*, 2015). Besides somatic expression (Fig EV2C), we localized secretagogin to both NE dendrites (Fig EV2C1) and axons (Figs EV2D and EV4B), including synaptosomes (Fig EV2F), suggesting that its interactome could modulate NE activity even at locations positioned distally from NE perikarya. We then used neuropeptide Y and enkephalin to mark subsets of TH$^+$ neurons (Finley *et al*, 1981; Everitt *et al*, 1984) to show that secretagogin labeled both NE subsets in the LC (Fig EV2F, enkephalin not shown). This suggests a functional relationship to TH itself rather than to a specific neuromodulatory peptide.

We sought to address causality between liquor-borne CNTF, secretagogin, and TH phosphorylation at its Ser$^{31}$ residue by acute icv administration of CNTF in freely moving *Cntf*$^{-/-}$, *Scgn*$^{-/-}$, and wild-type mice. We reasoned that *Cntf*$^{-/-}$ mice could respond like wild-type littermates with CNTFR signaling in NE neurons remaining signal competent. In turn, *Scgn* ablation could occlude CNTF-induced TH phosphorylation by disrupting the signal transduction machinery downstream from CNTFRs (Fig 4A). As such, we find Ser$^{31}$-phosphorylated TH in *Cntf*$^{-/-}$ at levels equivalent to those in wild-type mice (~1.5-fold increase; Figs 4C and EV2G). In contrast, CNTF infusion was ineffective to modulate TH phosphorylation in *Scgn*$^{-/-}$ mice (Fig 4C). These results provide genetic evidence for secretagogin linking CNTFR activation to TH phosphorylation in NE neurons.

To further dissect how secretagogin regulates TH phosphorylation, we first identified INS-1E cells (Merglen *et al*, 2004) as a cellular model in which CNTFR, TH, secretagogin, and Erk1 are natively co-expressed (Fig EV2H and H1). We then showed that RNAi-mediated *Scgn* silencing in INS-1E cells occludes CNTF-induced TH phosphorylation at Ser$^{31}$ (Figs 4D, and EV2I and J). Likewise, CNTF-induced Erk1 phosphorylation was abolished by *Scgn* knock-down (Fig 4E). Conspicuously, immunoprecipitation for secretagogin led to Erk1

co-precipitation in cranial pons homogenates (Fig 4F), suggesting that secretagogin could modulate TH phosphorylation through protein–protein interactions with Erk1. We then tested whether secretagogin regulates TH phosphorylation *in vivo* by subjecting *Scgn*$^{-/-}$ and wild-type littermate mice to formalin stress and assaying Ser$^{31}$ phosphorylation in their brainstem. Stress-induced TH phosphorylation did not occur in *Scgn*$^{-/-}$ mice ($P = 0.89$ control vs. stress) as opposed to their wild-type littermates (43% increase, $P < 0.05$ vs. *Scgn*$^{-/-}$ mice; Fig 4G) when, notably, *Scgn* ablation did not change total TH levels (Fig EV2K). In sum, our *in vivo* and *in vitro* data mechanistically place secretagogin into a hierarchical signal transduction cascade controlling TH phosphorylation in NE neurons.

If secretagogin$^+$ NE neurons that project to the PFC drive stress-induced defense, then their chemogenetic manipulation could manifest as freezing/hypolocomotion in a novel environment. We have tested this hypothesis by injecting AAV particles carrying Cre-dependent DREADD expression systems for neuronal activation (hM3Dq; Alexander *et al*, 2009) or inactivation (hM4Di; Armbruster *et al*, 2007) into the LC of *Scgn*-Cre mice (Fig 4H). Once placing CNO pre-treated animals (10 min; 2 mg/kg) into an open field, we find that chemogenetic NE activation induces freezing, rendering the animals persistently immobile (Figs 4H1 and EV3A; Movies EV1–EV3). In contrast, hM4Di-treated animals remained mobile (Fig 4H1). We then repeated this experiment by exposing subsets of mice to formalin stress and CNO infusion 15–20 min prior to recordings. Here, hM3Dq-treated animals that underwent formalin stress exhibited a more uniform freezing response (Fig EV3A and A1) suggesting the formalin-induced reinforcement of CNO action. By showing that hM3Dq-carrier mice lacked an anxiogenic response when subjected to an elevated plus maze task (Harkany *et al*, 1999; Fig EV3B), we excluded that changes in open-field activity were due to anxiety.

## *Scgn*$^{-/-}$ mice are resilient to acute stress

Next, we argued that *Scgn*$^{-/-}$ mice could be resilient to acute stress because of their blunted NE production and, consequently, NE inactivity in PFC when exposed to stress acutely. Considering that NE inputs to the mPFC drive behavioral flexibility (Uematsu *et al*, 2017), we opted for behavioral paradigms that reveal sensory components of stress sensitivity without being reliant on enforcement or association (Tovote *et al*, 2015). Since secretagogin is expressed in sensory dorsal root ganglia (Shi *et al*, 2012a), we first verified that *Scgn*$^{-/-}$ mice had unchanged sensory thresholds in the von Frey filament test, as compared to wild-type littermates ($P = 0.19$; Fig EV3C). Thereafter, we showed that *Scgn*$^{-/-}$ mice produce reduced escape responses when triggering their pinna reflex (Fig EV3C1) or exposing them to toe pinch ($P < 0.05$ vs. wild-type littermates; Fig EV3C2). Additionally, *Scgn*$^{-/-}$ mice do not bite when provoked [$0.33 \pm 0.17$ (wild-type) vs. $0.92 \pm 0.21$ (*Scgn*$^{-/-}$, biting index, $P < 0.05$)]. Lastly, *Scgn*$^{-/-}$ mice habituate to a novel environment indistinguishable from wild-type controls (Fig EV3D and D1). Nevertheless, when delivering a random sequence of unavoidable foot-shocks, their period of immobility ("freezing") is significantly reduced as compared to wild-type littermates ($P < 0.05$; Fig EV3D1). Since neither the distance nor the pattern of their mobility differs significantly from control mice (Fig EV3D and D2), we conclude that genetic ablation of *Scgn* could impede NE network determinants of stress-associated behaviors to produce resilience.

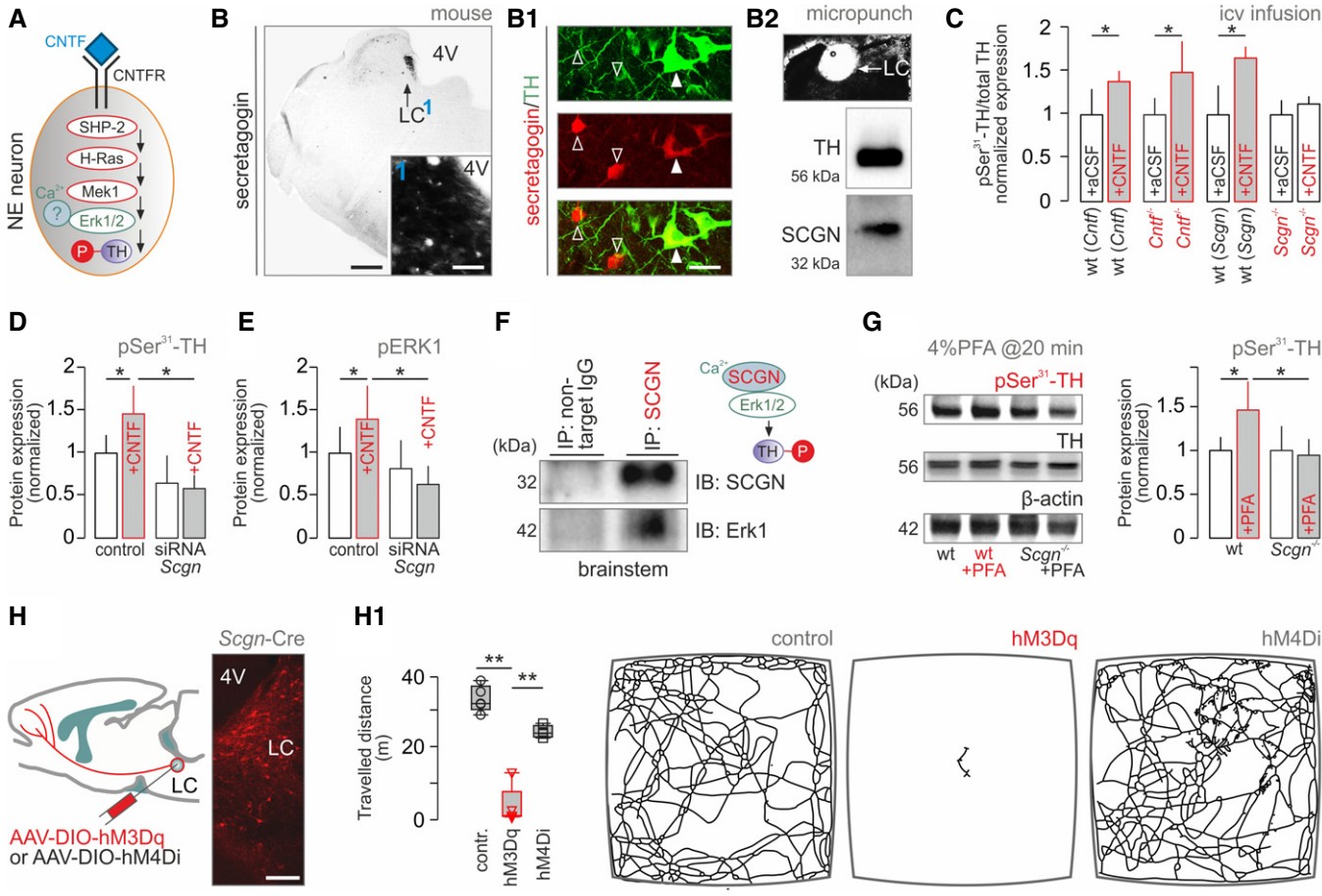

**Figure 4.  Secretagogin recruits Erk1 for tyrosine hydroxylase phosphorylation in mouse noradrenergic neurons.**

A   Graphical rendering of the proposed intracellular signaling cascade based on reports placing Erk1 as a function determinant for tyrosine hydroxylase (TH) by phosphorylation (Tekin *et al*, 2014).

B   Histochemical detection of secretagogin (SCGN) in mouse LC. (1) depicts the dense meshwork of local fibers, likely dendrites, emanating toward the 4th ventricle (4V). (B1) Secretagogin co-localizes with TH in many (*solid arrowheads*) but not all (*open arrowheads*) neurons in the LC. (B2) Micropunches of native tissue followed by Western analysis confirm the presence of both TH and SCGN in the LC. Scale bars = 100 μm (B), 30 μm (B inset), and 10 μm (B1).

C   TH phosphorylation upon acute CNTF infusion into the 3rd ventricle in freely moving $Cntf^{-/-}$, $Scgn^{-/-}$ and wild-type mice. n = 10/group; *$P < 0.05$ vs. wild-type. Data were normalized to controls not exposed to formalin stress. Representative Western blots are shown in Fig EV2G.

D   Recombinant CNTF induces TH phosphorylation at Ser31 in INS-1E cells, which co-express SCGN, Erk1, and TH. RNAi silencing of *Scgn* abolishes TH activation (from triplicate experiments; *$P < 0.05$).

E   Recombinant CNTF also induces Erk1 phosphorylation in a *Scgn*-dependent manner in INS-1E cells (from triplicate experiments; *$P < 0.05$).

F   Immunoprecipitation (IP) using an anti-secretagogin antibody as bait led to the co-elution of Erk1 in mouse brainstem.

G   Formalin stress increases Ser31-phosphorylated TH levels in the LC of wild-type but not $Scgn^{-/-}$ mice. n = 5/genotype; *$P < 0.05$.

H   Microinjection of AAV particles carrying activating (hM3Dq) and inactivating (hM4Di) DREADD constructs in *Scgn*-Cre mice. Histochemistry shows site-directed mCherry expression in the LC 14–28 days later. Scale bar = 250 μm. (H1) *Left*: CNO-induced freezing in hM3Dq but not hM4Di carriers. Data in box plots represent medians and 10th, 25th, 75th, and 90th percentiles. **$P < 0.01$ vs. control or hM4Di, $n \geq 4$ animals/group. *Right*: Representative trajectories in the open field during 5 min.

Data information: Data are expressed as means ± s.e.m. and were statistically analyzed by either Student's *t*-test (C, D, E, G) or ANOVA (general linear model; H1).

## Severe pathogenic stress induces TH phosphorylation in human LC

If stress-induced TH phosphorylation is of pathophysiological significance, then it might be detected in humans with severe acute stress (and associated pain). In support of this hypothesis, we find increased Ser31 phosphorylation *post-mortem* (with concomitant increases in TH content itself) in LC micropunches (Palkovits, 1973) in suicide subjects (108 and 44% increase in native and pSer31-phosphorylated TH levels, respectively, $P < 0.05$; Figs 5B

and EV4A), as well as those succumbing to acute heart failure (150 and 129% increase in total and Ser31-phosphorylated TH levels relative to controls, respectively, $P < 0.05$; Fig 5C and C1). Moreover, when performing immunoprecipitation with secretagogin as bait in *post-mortem* LC specimens of subjects with acute heart failure, we find substantially higher Erk1 content in the eluent fraction, as compared to control subjects and non-targeting IgG controls (Fig 5D). In sum, these data suggest that stress-induced NE sensitization is a mechanism conserved among mammalian species, including humans.

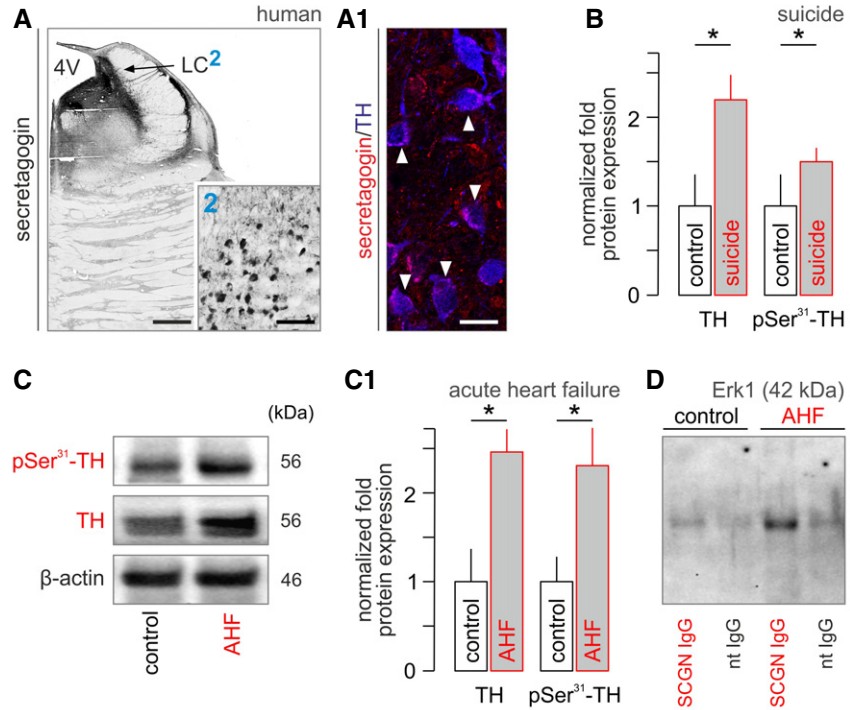

**Figure 5.   Tyrosine hydroxylase phosphorylation in humans experiencing severe stress.**

A   The LC contains secretagogin[+] neurons in humans, as well. (2) depicts secretagogin-labeled perikarya residing in a dense meshwork of local fibers, likely dendrites. (A1) Secretagogin co-exists with TH in most neurons *(solid arrowheads)*. Scale bars = 350 μm (A), 120 μm (A/2), and 12 μm (A1).

B   Total TH levels and their phosphorylation at Ser[31] in suicidal subjects relative to age-matched controls [$n$ = 6 (suicide) and $n$ = 7 (controls)]. *$P$ < 0.05. Representative Western blots are shown in Fig EV4A.

C   Total and Ser[31]-phosphorylated TH in acute heart failure (AHF) in human subjects. Representative Western blot. (C1) Data from $n$ = 7 subjects with acute heart failure and $n$ = 6 controls. *$P$ < 0.05.

D   Immunoprecipitation (IP) using an anti-secretagogin antibody as bait revealed increased secretagogin (SCGN)-Erk1 interaction in LC micropunches after AHF. nt IgG, non-targeting IgG. Representative immunoblot is shown. Experiments were performed in triplicate.

Data information: Data are expressed as means ± s.e.m. and were analyzed by Student's *t*-test (B, C).

## Secretagogin controls NE-induced neuronal excitability in the prefrontal cortex

Brainstem NE neurons regulate alertness and emotional flexibility (Uematsu *et al*, 2017) through their direct efferents to the mPFC (Fig 6A), which is identified as the cortical "hub" for escape behaviors (McNaughton & Corr, 2004; Wu *et al*, 2016; Schwabe, 2017). Therefore, and based on our experimental results, we posited that the mPFC is the final element of a hypothalamus-LC-PFC tripartite circuit that drives stress-induced behaviors. In mice, we focused on the infra- and prelimbic cortices, which correspond to the ventromedial area of the mPFC in humans and are connected to the hypothalamus and amygdala to regulate emotional actions (Arnsten, 2009). First, we used BDA-based anterograde tract tracing (Fig 6A) to show that TH[+] efferents to the mPFC contain Ser[31]-phosphorylated TH (Fig 6B and B1), which is typically present in *en-passant* boutons. For secretagogin to modulate TH activity in NE axons, it ought to be co-expressed with pSer[31]-TH in the mPFC. Indeed, Western analysis suggests that secretagogin–TH regulatory interactions could dominate in the mPFC (Fig 6C, more so than in, e.g., somatosensory areas) with its presence in synaptosomal fractions reinforcing presynaptic modulation (Fig EV2F). Moreover, *Scgn*[−/−] mice failed to

display Ser[31] phosphorylation of TH in the mPFC relative to wild-type littermates [−18% (*Scgn*[−/−]) vs. +30% (wild-type), $P$ < 0.05; Fig 6D and D1]. Since stress-induced TH activation by phosphorylation supports NE synthesis *in loco*, we hypothesized that NE signaling is chronically impaired in *Scgn*[−/−] mice and the lack of a stress-induced NE surge precludes stress-induced behavioral phenotypes. To this end, we determined cortical NE content in *Scgn*[−/−] and wild-type mice exposed to acute stress. Indeed, peripheral formalin injection increased NE production in the mPFC of wild-type animals [0.25 ± 0.02 (control) vs. 0.31 ± 0.02 (stress) NE μg/mg tissue at 120 min after stress induction (Fig 6E)]. In turn, *Scgn*[−/−] mice failed to produce a stress-induced NE surge [0.28 ± 0.02 (control) vs. 0.27 ± 0.03 NE μg/mg tissue, $P$ = 0.73; Fig 6E]. These changes did not affect, e.g., the somatosensory cortex and only perturbed NE but neither dopamine nor 3,4-dihydroxyphenylacetic acid (DOPAC) levels (Fig EV5H). These data suggest that NE production is impaired in the mPFC of *Scgn*[−/−] mice, which could underpin behavioral insensitivity to acute stress.

If the lack of NE inputs to the mPFC of *Scgn*[−/−] mice is functionally significant, then altered excitability of pyramidal neurons might be expected to ensue. *Scgn*[−/−] layer 5 pyramidal cells exhibited significantly hyperpolarized resting membrane potentials

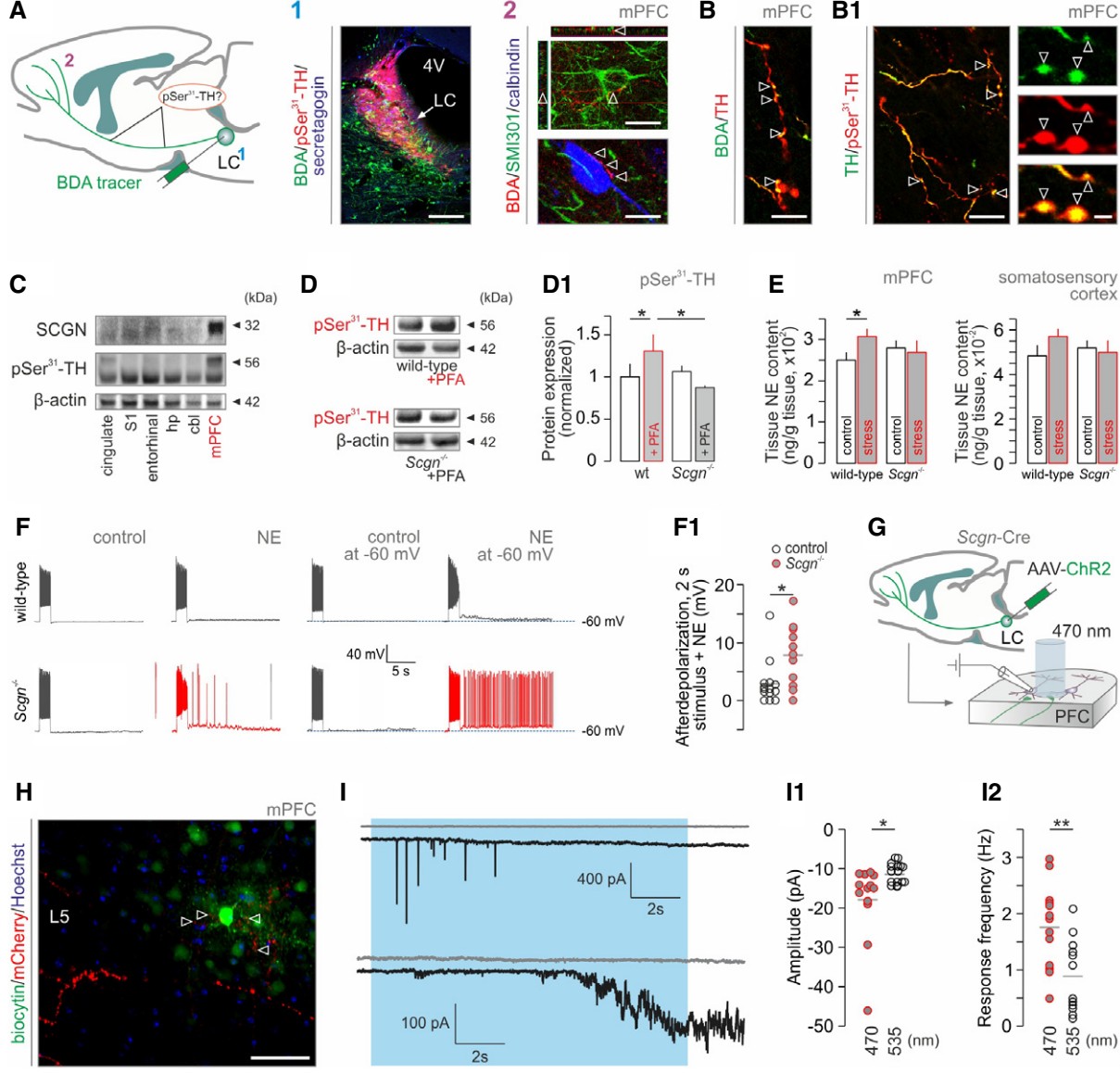

**Figure 6. Secretagogin gates noradrenergic activity in PFC.**

A   Schema of noradrenergic projections to the medial prefrontal cortex (mPFC) used for tract tracing and the analysis of axonal tyrosine hydroxylase (TH) phosphorylation. (1) Biotinylated dextran amine (BDA) injected into the mouse LC led to (2) BDA-labeled TH⁺ axons contacting pyramidal cells (*upper*) and interneurons (*lower*) in the mPFC (*open arrowheads*). Scale bars = 100 µm (1) and 10 µm (2).

B   TH⁺ axons (*open arrowheads*, B) contained TH phosphorylated at Ser31 (*open arrowheads*, B1). Scale bars = 10 µm (B, B1/left) and 2 µm (B1/right).

C   Secretagogin (SCGN) is abundantly co-expressed with Ser31-phosphorylated TH in mPFC (note marginal expression in somatosensory cortex). cbl, cerebellum; hp, hippocampus; mPFC, medial prefrontal cortex; S1, primary somatosensory cortex.

D   Formalin stress induces TH phosphorylation in the mPFC of wild-type but not $Scgn^{-/-}$ mice; $n$ = 5/genotype; *$P$ < 0.05.

E   $Scgn^{-/-}$ mice fail to produce NE in mPFC upon acute stress ($n \geq 4$/group). Figure EV5H is referred to for data on unchanged dopamine and DOPAC levels. Note that acute formalin stress failed to significantly elevate NE in the somatosensory cortex ($P$ = 0.259) even if a differential response to stress between wild-type and $Scgn^{-/-}$ mice is noticeable. *$P$ < 0.05.

F   NE superfusion *ex vivo* increases the excitability of layer (L) 5 pyramidal cells in the mPFC of $Scgn^{-/-}$ mice relative to wild-type littermates. (F1) Quantitative data. *$P$ < 0.05, $n \geq 3$ mice/genotype.

G   Experimental protocol to test whether optogenetic activation of NE efferents to the mPFC is sufficient to increase cortical network excitability. Fourteen days after AAV infusion, experiments were performed in acute brain slices spanning the mPFC using 470 and 535 nm (control) excitation wavelengths.

H   Biocytin-filled L5 pyramidal neuron receives channelrhodopsin-2/mCherry-containing NE inputs. *Open arrowheads* point to varicosities apposing pyramidal dendrites. Scale bar = 100 µm.

I   Representative images of heightened glutamatergic responses in L5 pyramidal cells upon 470-nm light excitation (blue shading). Gray traces (top) show the lack of excitability when exposed to light at 535 nm. (I1) Amplitude of postsynaptic currents at both wavelengths. *$P$ < 0.05 vs. control at 535 nm. (I2) Light-induced response frequencies. **$P$ < 0.01 vs. control at 535 nm.

Data information: Data are expressed as means ± s.e.m. and were statistically tested by Student's $t$-test (D1, E) or one-way ANOVA (F1, I1, I2).

(Fig EV5A–C) and lowered voltage thresholds to generate action potentials (APs; Fig EV5D–D2) *ex vivo*, suggesting their increased excitability. We also found that $Scgn^{-/-}$ pyramidal cells are increasingly sensitive to NE (10 μM) applied exogenously [2.5 ± 0.7 mV (wild-type) vs. 8.1 ± 2.1 ($Scgn^{-/-}$), $P < 0.05$; Figs 6F and F1, and EV5E–G], which we interpret as network sensitization to, and resetting of inherent flexibility toward the lifelong lack of tonic NE triggers. This notion is supported by marked increases in afterdepolarization effects on electrical self-stimulation in the presence of NE (Figs 6F and EV5G). We then employed an optogenetic approach to confirm that the activation of secretagogin-expressing NE neurons in the LC is sufficient to reset mPFC network excitability (Fig 6G). By introducing channelrhodopsin into NE neurons in *Scgn*-Cre mice, we labeled PFC projections 14–17 days later (Fig 6H). Since NE does not induce electrogenic currents *per se* but instead lowers the excitation threshold for glutamate through metabotropic α-adrenergic receptors (Ramos & Arnsten, 2007; Santana *et al*, 2013), single short-lived excitation pulses were without effect (data not shown). However, when using 30-s laser excitation at 470 nm (but not 535 nm), repetitive high-amplitude synaptic currents (9/15 cells; Fig 6I, *upper trace*) or activation of an inward current in the 30–150 pA range (3/15 cells; Fig 6I, *lower trace*) was recorded in layer 5 pyramidal cells (Fig 6I–I2). No such event was recorded under non-stimulated conditions. These data show that secretagogin selectively co-exists in the mPFC where it can control NE release onto pyramidal cells to modulate network output. Since NE neurotransmission is increased in stress, mediates cortical excitability in chronic states (Arnsten, 2009), and underpins fight/flight behaviors (Bremner *et al*, 2008), our results uncover a multimodal pathway that combines volume and synaptic neurotransmission to modulate excitatory output commands from the PFC.

## Discussion

Collectively, we demonstrate the existence of a multimodal signaling pathway including both synaptic signaling and volume transmission from ependymal cells via the cerebrospinal fluid to maintain NE-induced cortical excitability over prolonged periods. This pathway bypasses classical anatomical units of ascending sensory pathways (spinal, thalamic, and somatosensory cortical nodes) to express post-traumatic vigilance. At the molecular level, and alike the HPA axis, CRH (Swanson & Sawchenko, 1980), secretagogin (Romanov *et al*, 2015), and NE (Korf *et al*, 1973) are used as mediators yet through the recruitment of novel cellular interactions for effective modulation of mPFC output. Moreover, our results suggest that CRH neurons might simultaneously orchestrate HPA activation and cortical excitability, which can be an advantage to overcome metabolic restrictions at the periphery.

Acute stress triggers a complex set of behaviors, which substantially differ in their magnitude and repertoire dependent on the context and duration. Consequently, substantial variations have been recorded in the extent, temporal dynamics, and connectivity diagrams of the neuronal circuits that underpin responses to stressful stimuli; e.g., hunger and fasting (but also overfeeding) activates proopiomelanocortin, neuropeptide Y, and glutamate (VGLUT2$^+$) neurons in the arcuate nucleus of the hypothalamus through circulating hormones (Sternson *et al*, 2005; Campbell *et al*, 2017; Fenselau *et al*, 2017), and aggression toward (and defense against) an intruder activates the ventromedial hypothalamus (Lin *et al*, 2011), while maternal behaviors are executed by galanin-containing neurons of the medial preoptic area (Kohl *et al*, 2018). Regardless of the cellular identity of the primary responder network, stress-related command is executed by CRH neurons of the PVN, which are activated by ascending glutamatergic pathways (Fenselau *et al*, 2017). Undoubtedly, CRH (and likely glutamate) release at the median eminence is the dominant function for CRH neurons to mobilize glucocorticoids to set bodily (particularly muscular) "fight/flight" responses. Nevertheless, a key postulate of our present and earlier (Romanov *et al*, 2017a) studies is that molecularly heterogeneous CRH neurons shape more than a single output pathway, with glutamatergic CRH$^+$ neurons integrating and propagating the most diverse hypothalamic network activity into a binary code. Once activated, ependymal cells convert synaptic inputs into a long-lasting metabolic state change with CNTF released into the 3$^{rd}$ ventricle. This is important since it not only links PVN output to distal midbrain effector systems (such as NE efferents) but also suggests that this event ensures context independence: NE activation occurs under the most diverse stress conditions (Arnsten, 2009). This understanding also mandated our approach to assess stress-related behaviors. Irrespective of the threat, acute stress invariably triggers an initial freezing response before context-dependent avoidance behavior manifests (Korte, 2001; Swiergiel *et al*, 2007, 2008; Egan *et al*, 2009). Therefore, we focused on this critical period of immobility with its duration interpreted as a behavioral correlate of the extent of NE afferent activation in the PFC. We show at successive levels of the PVN-LC-PFC circuit (Fig 1A) that manipulation of CNTF release in ventricular ependyma, secretagogin-dependent NE production in LC, or glutamate signaling in the PFC lengthens freezing behavior. Thus, we uncover a causal neural determinant that drives the initial behavioral stage of the stress response, which then can be followed by a refined repertoire of conscious and/or metabolic decisions to ensure adequate and proportionate response to a particular environmental stressor.

CRH$^+$ parvocellular cells of the PVN innervate the median eminence to release their neuropeptide content into the portal circulation for stimulated ACTH release. Recently, we have demonstrated that both glutamatergic and GABAergic subtypes of CRH$^+$ neurons exist in the mammalian hypothalamus (Romanov *et al*, 2015, 2017b). We have recently tied CRH$^+$ neuroendocrine cells to the HPA axis as its first-responder "*stress-on*" components by identifying their use of *Scgn*, a parvocellular neuron-specific Ca$^{2+}$-sensor protein, to facilitate fast vesicular neuropeptide secretion (Romanov *et al*, 2015, 2017a). However, the classical concept of CRH release leaves the fast neurotransmitter content (irrespective of that being glutamate or GABA) unaccounted for. Here, we combine viral circuit mapping, high-resolution morphological analysis of GFP-tagged synapses formed by CRH$^+$ neurons, single-cell RNA-seq of their ependymal targets, and chemical probing to suggest the predominant use of glutamate for the focal stimulation of CNTF release. Our interpretation is supported by the lack of CRH receptors mRNAs (*Crhr1/Crhr2*) in ependymal cells together with frequent VGLUT2 and VGLUT1 co-localization in CRH$^+$/GFP$^+$ presynapses. Thus, we have successfully exploited the power of "forward transcriptomics" (Romanov *et al*, 2015) to predict molecular circuit determinants, including the identity of the effector molecule released by ependymal cells in an activity-dependent fashion.

Mono- or polysynaptic neurotransmission is the most precise form of intercellular communication in the brain. The "tripartite synapse doctrine" includes astroglial end-feet (Araque et al, 1999) as well as efficient re-uptake mechanisms to eliminate surplus neurotransmitters, thus spatially confining their action and limiting spillover (Huang & Bordey, 2004). In contrast, to modulate brain functions, the action of blood-borne peripheral messengers, such as peptide hormones and lipids, is specified by their ability to cross the blood–brain barrier and the cell-type specificity of their cognate receptors (Alpár & Harkany, 2013). An intermediary between these mechanisms is volume transmission, defined as a lack of specific and spatially demarcated conduits between the site of ligand production and receptor-mediated signaling within a specific tissue (Fuxe et al, 2013). Purines and purine nucleotides (e.g., ATP), catecholamines, and hormones are commonly used for both short- and long-range volume transmission (Fuxe et al, 1988; Roper, 2007; Housley et al, 2009). Here, we define a novel function for a developmentally regulated neurotrophin (Sendtner et al, 1994) by showing stress-induced CNTF release into the cerebrospinal fluid in the adult brain. This finding is significant since it allows the conversion of fast synaptic excitation ("bouts of activity") into long-lived neuromodulation through the use of the brain's aqueductal system. Our results also strengthen the recent concept that most signaling systems for axon growth and directional guidance including brain-derived neurotrophic factor (BDNF)–tropomyosin receptor kinase B (TrkB; Song et al, 1997), Slit–Roundabout (Alpár et al, 2014; Guan et al, 2007), endocannabinoid (Berghuis et al, 2007), and semaphorin–plexin signaling (Orr et al, 2017) switch from their developmental roles to tune synaptic neurotransmission and plasticity, often via retrograde signaling, in the postnatal brain. Here, CNTF satisfies a role to enhance NE activity in the LC, alike BDNF released onto cortical interneurons (Inagaki et al, 2008; Magby et al, 2006), but does so through volume transmission in the liquor space. Then, CNTF action is specified by the expression of its cognate Trk receptor (CNTFRα) in NE neurons of the brainstem (Ip et al, 1993). It is noteworthy that Trk receptor activation assures fast onset (within 1–2 min) with signaling up to hours depending on the signal transduction pathway (Sermasi et al, 2000; Mizoguchi et al, 2002; Rosch et al, 2005). Thereby, Trk receptors might be best suited to produce activity patterns that match the species' needs for survival and produce evolutionary advantage.

CNTF is only one neuromodulator whose mRNA is found in ependymal cells (Fig 1C). Single-cell RNA-seq also resolved gene expression for BDNF, vgf, which produces alternative peptide products for release in endocrine tissues (Ferri & Possenti, 1996), and neuropeptides, suggesting their combinatorial action if co-released. Alternatively, mature peptides might be partitioned to spatially segregated subsets of ependymal cells entrained by distinct neuronal inputs for function determination, or their release depends on specific biophysical properties to gain specificity. Even if these neuroactive substances were ubiquitously co-released, then the receptor repertoire of their cellular targets could diversify their action. Ultimately, ependymal cells could thus produce temporary neuropeptide constellations for volume transmission whose differential binding and output modulation in nearby midbrain and hindbrain structures could account for substantial functional heterogeneity and explain key features of neuropeptide contributions to anxiety and depression (Belzung et al, 2006; Barde et al, 2016).

NE neurons are commonly identified by their ubiquitous TH expression. Recently, functional evidence together with viral circuit tracing and optogenetic analysis demonstrates that a subset of NE neurons specifically innervates the PFC to drive emotional flexibility, while others project to the amygdala to modulate fear (Uematsu et al, 2017). Here, our data suggest that the co-existence of CNTFRα, secretagogin, and TH is minimally sufficient to distinguish PFC-projecting NE neurons even without making use of their deeper transcriptome. Molecular characteristics of NE neurons in humans suggest evolutionary conservation of this circuit, which is supported by clinicopathological data from stratified cohorts of humans with acute heart attack-induced pain, which is considered as the strongest stressor known to humans. Notably, our animal models confirm that breaking hypothalamus-LC-mPFC signaling at the level of secretagogin in NE neurons renders animals insensitive to pain-induced stress. Thus, our findings are not only significant in the context of psychiatric disorders (e.g., post-traumatic stress disorder) but extend to a wider spectrum of states, including nosocomial interventions (e.g., patients who show oversensitivity to dental intervention-induced pain; Klepac et al, 1980). Considering that the maintenance of heightened NE activity in the cerebral cortex characterizes the transition from acute to chronic stress (Arnsten, 2009), our findings might gain importance as a druggable cellular platform (e.g., CNTFRα inhibitors or modulators of $Ca^{2+}$-sensor proteins) to combat devastating human neuropsychiatric conditions (Bremner et al, 2008).

# Materials and Methods

### Reagents and tools table

| Reagent/resource | Source | Identifier |
|---|---|---|
| **Antibodies** | | |
| β-Actin | Sigma | Cat#A1978 |
| Calbindin | Synaptic Systems | Cat#2014104 |
| CNTF | R&D Systems | Cat#AF-557-NA |
| CNTF receptor | R&D Systems | Cat#AF-559-NA |
| CRH | Santa Cruz | Cat#SC-1759 |
| Enkephalin | Immunostar | Cat#20065 |

**Reagents and tools table**  (continued)

| Reagent/resource | Source | Identifier |
| --- | --- | --- |
| Erk1/2 | Cell Signaling | Cat#9107 |
| Phosphorylated Erk1/2 | Cell Signaling | Cat#9101 |
| GFAP | DAKO | Cat#Z0334 |
| GRIA1 | Alomone | AGP-009 |
| Neuropeptide Y | Immunostar | Cat#22940 |
| Nestin | Millipore | MAB353 |
| Secretagogin | Gift from L. Wagner | (rabbit) |
| Secretagogin | R&D Systems | Cat#AF4878 |
| Synaptophysin | Synaptic Systems | Cat#101-002 |
| SMI301 | BioLegend | Cat#801702 |
| Tyrosine hydroxylase (TH) | Merck/Millipore | Cat#AB152 |
| TH phosphorylated at Ser[31] | Sigma | Cat#SAB4300674 |
| TH phosphorylated at Ser[40] | Sigma | Cat#T9573 |
| VGLUT1 | Synaptic Systems | Cat#135-202 |
| VGLUT2 | Synaptic Systems | Cat#135-404 |
| Biotinylated Donkey Anti-Rabbit IgG | Jackson ImmunoR. | Cat#711-065-152 |
| Cy2-conjugated Donkey Anti-Goat IgG | Jackson ImmunoR. | Cat#705-225-003 |
| Cy2-conjugated Donkey Anti-Rabbit IgG | Jackson ImmunoR. | Cat#711-225-152 |
| Cy3-conjugated Donkey Anti-Goat IgG | Jackson ImmunoR. | Cat#705-165-003 |
| Cy3-conjugated Donkey Anti-Rabbit IgG | Jackson ImmunoR. | Cat#711-165-152 |
| Cy3-conjugated Donkey Anti-Guinea Pig IgG | Jackson ImmunoR. | Cat#706-165-148 |
| Cy5-conjugated Donkey Anti-Rabbit IgG | Jackson ImmunoR. | Cat#711-175-152 |
| Cy2-Streptavidin | Jackson ImmunoR. | Cat#016-220-084 |
| Normal Goat IgG | Santa Cruz | Cat#sc-2028 |
| Dynabeads Protein G | Invitrogen | Cat#10004D |
| Donkey Anti-Rabbit HRP | Jackson ImmunoR. | Cat#711-035-152 |
| Donkey Anti-Mouse HRP | Jackson ImmunoR. | Cat#715-035-151 |
| Donkey Anti-Goat HRP | Jackson ImmunoR. | Cat#705-035-147 |
| **Chemicals, peptides, and recombinant proteins** | | |
| L-norepinephrine | Sigma | Cat#74480 |
| Recombinant Ciliary Neurotrophic Factor | Sigma | Cat#C3710 |
| Biotinylated Dextran Amine (BDA, 10,000 MW) | Invitrogen | Cat#D1956 |
| Biotinylated Dextran Amine (BDA, 3,000 MW) | Molecular Probes | Cat#D7135 |
| Clarity Western ECL Substrate | BIO-RAD | Cat#170-5060 |
| Vectastain ABC Kit | Vector Labs | Cat#PK-6101 |
| DAB Peroxidase Substrate Kit | Vector Labs | Cat#SK-4100 |
| jetPRIME | Polyplus | Cat#114-15 |
| TSA Plus Fluorescein System Kit | PerkinElmer | NEL741001KT |
| Ketamine (Calypsol) | Richter | Cat#A6A066 |
| Xylazine (Nerfasin) | Le Vet Pharma | Cat#15H044 |
| Isoflurane (Forane®) | AbbVie GmbH | Cat#B506 |
| Hoechst 33,342 | Thermo Fisher | Cat#H1399 |
| **Critical commercial assays** | | |
| CNTF Rat ELISA kit | Thermo Fisher Sci. | Cat#ERCNTF |

**Reagents and tools table**  (continued)

| Reagent/resource | Source | Identifier |
|---|---|---|
| **Deposited data** | | |
| Single-cell RNA-seq of mouse hypothalamus cells from control and acute formalin-stressed animals | NCBI Gene Expression Omnibus | GSE74672 |
| **Experimental models: cell lines** | | |
| INS-1E | Gift from P. Maechler | |
| Experimental models: organisms/strains | | |
| *Crh-Ires*-Cre::egfp mouse | Gift from J.S. Bains | Cat#012704 |
| *cfos*-Cre$^{ERT2}$ mouse | JAX Labs | Cat#021882 |
| *ROSA26-stop-ZsGreen1$^{f/f}$* mouse (*Ai6*) | JAX Labs | Cat#007906 |
| *Scgn$^{-/-}$* mice | MMRRC for authors | Malenczyk *et al* (2017) |
| *Cntf$^{-/-}$* mice | A. Giordano | Masu *et al* (1993) |
| *Scgn*-Cre mouse | G. Szabó | Developed for this study |
| AAV8-EF1a-double floxed-hChR2(H134R)-mCherry-WPRE-HGHpA | Addgene Viral Service | Cat#20297 |
| AAV8-hSyn-DIO-hM3D(Gq)-mCherry | Addgene Viral Service | Cat#44361 |
| AAV8-hSyn-DIO-hM4D(Gi)-mCherry | Addgene Viral Service | Cat#44362 |
| AAV8-hSyn-DIO-hM3D(Gq)-mCherry | Addgene Viral Service | Cat#44361 |
| pAAV-hSyn-DIO-mCherry | Addgene Viral Service | Cat#50459 |
| AAV-hSyn1-GCaMP6f-P2A-nls-dTomato | Addgene Viral Service | Cat#51085 |
| **Sequence-based reagents** | | |
| Primers for CNTF | IDT | Ref#72797892-3 |
| Control siRNA | Santa Cruz | Cat#sc-37007 |
| SCGN siRNA | Santa Cruz | Cat#sc-153255 |
| Accell SMARTpool CNTF siRNA | GE Dharmacon | Cat#A-091086 |
| Accell SMARTpool SCGN siRNA | GE Dharmacon | Cat#A-092169 |
| Accell SMARTpool non-targeting siRNA | GE Dharmacon | Cat#D-001950-01 |
| **Software and algorithms** | | |
| Clampfit 10.0 | Molecular Devices | Ver. pCLAMP-10 |
| MiniAnalysis | Synaptosoft | "MiniAnalysis" |
| SigmaPlot | Systat | |
| Statistical Package for the Social Sciences 17.0 | SPSS Inc | |
| Image Lab 5.0 | BIO-RAD | |

## Methods and Protocols

### Animals

*Scgn$^{-/-}$* mice were custom-generated at MMRRC (Mouse Biology Program, University of California) using the "two-in-one" targeting strategy (Skarnes *et al*, 2011), which generates full knock-outs by expressing a termination signal after exon 3 of the secretagogin gene. The ensuing truncated protein terminates before the first EF-hand domain, excluding Ca$^{2+}$-binding activity. *Crh-Ires*-Cre (from J.S. Bains, University of Calgary, Canada) were crossed with B6.Cg-*Gt(ROSA)26Sor$^{tm6(CAG-ZsGreen1)Hze}$*/J mice (*Ai6*; JAX stock #012704 and #007906) to visualize periventricular innervation by CRH neurons (Romanov *et al*, 2017a,b). *Scgn*-Cre mice were developed using the BAC technology (Calvigioni *et al*, 2017; Z.M., F.E., and G.S., Institute of Experimental Medicine, Hungarian Academy of Sciences). When using Wistar rats, experiments were performed

at 12 weeks of age. Animals were kept under standard housing condition (including a 12-h/12-h light/dark cycle) with food and water available *ad libitum*. Experimental procedures, including CSF sampling from the 4$^{th}$ ventricle and transcardial perfusion, were approved by the ethical review boards of the Semmelweis University (PE/EA/1234-3/2017, Hungary) and the Medical University of Vienna (BMWFW-66.009/0277-WF/V/3b/2017, Tierversuchsgesetz 2012, BGBI, Nr. 114/2012), including the size estimate of each experimental cohort. All procedures conformed to the European Convention for the Protection of Vertebrate Animals used for experimental and other scientific purposes (86/609/EEC). Wherever possible, animals of the same genotype were randomly assigned to experimental manipulations (control vs. various treatments) to reduce procedural bias. During surgery or transcardial perfusion, animals were anesthetized intramuscularly (i.m.) or intraperitoneally (i.p.) with a mixture of ketamine (50 mg/kg

b.wt.) and xylazine (4 mg/kg b.wt.) or by inhalation of isoflurane (at 5% with 1 l/min flow rate of tubed air).

### Locus coeruleus explants

Postnatal day (P) 5 rat brains ($n = 6$) were vibratome-sectioned coronally at 300 μm thickness in ice-cold DMEM containing penicillin (100 U/ml) and streptomycin (100 μg/ml, both from Invitrogen). *Ex vivo* slices containing the locus coeruleus (LC) were mounted on Millicell-CM culture inserts (0.4 μm pore size; Millipore) and equilibrated in Neurobasal Medium containing 2 mM GlutaMAX and 10% fetal bovine serum (FBS) for 2 h. Subsequently, the medium was replaced with Neurobasal Medium, GlutaMAX (2 mM), and B27 supplement (2%) for 24 h when recombinant CNTF (100 ng in 2.5 μl medium; Sigma) was applied directly onto each brain slice for 20 min. Tissues were then collected, washed, homogenized in TNE lysis buffer, and processed for Western blotting.

### Cell lines

Rat-derived INS-1E insulinoma cells (gift from P. Maechler; Merglen *et al*, 2004) were maintained in RPMI-1640 containing HEPES (10 mM), fetal bovine serum (FBS; 5%), Na-pyruvate (1 mM), GlutaMAX (2 mM), β-mercaptoethanol (50 μM), penicillin (100 U/ml), and streptomycin (100 μg/ml; all from Invitrogen). Cells were enzymatically dissociated and plated at a density of 500,000 cells/well on poly-D-lysine-coated coverslips in 6-well plates for Western blotting or qPCR.

### Human samples

For Western analysis, micropunched LC samples were obtained from subjects deceased due to acute heart failure or suicide at the Human Brain Tissue Bank and Laboratory, Semmelweis University, Hungary (Appendix Table S1). Tissues were obtained and used in compliance with the Declaration of Helsinki and following relevant institutional guidelines [Regional and Institutional Committee of Science and Research Ethics of Semmelweis University (TUKEB 84/2014)]. All patient material was coded to ensure anonymity throughout tissue processing.

### Formalin stress

Twelve-week-old animals were used in all experiments. Rats ($n = 8$) received an injection of 4% paraformaldehyde (PFA; in 50 μl physiological saline) into their right hind paw and subsequently returned to their home cages. After either 20 or 120 min, the animals ($n = 4$/time point) were decapitated and their brains dissected out. The medullary–pontine brainstem and medial prefrontal cortex were isolated and subjected to protein analysis by Western blotting. Control animals ($n = 4$) were removed from their home cages only for decapitation and tissue sampling. $Scgn^{-/-}$ mice and their wild-type littermates ($n = 3$ each) were subjected to an identical treatment but with subsequent tissue collection restricted to 20 min post-induction. Control mice of both genotypes were left in their home cages until decapitation to minimize bias by acute handling and immobilization. To discriminate the pool of stress-activated periventricular ependymal cells, $cfos$-Cre$^{ERT2}$::*ROSA26-stop-ZsGreen1$^{f/f}$* "TRAP mice" were used (JAX stock #021882 and #007906) as per published protocols (Guenthner *et al*, 2013). In brief, 24 h after a single-bolus tamoxifen injection (150 mg/kg), 4% PFA (in 50 μl

physiological saline) was injected as above to induce stress acutely. After a lag-time of 72 h to allow for ZsGreen expression in "stress-responder" cells, animals were transcardially perfused with a fixative containing 4% PFA in 0.1 M phosphate buffer (PB, pH 7.4) for quantitative histochemical analysis (see below).

### Virus microinjections in vivo

Stereotaxic injections were performed as described (Romanov *et al*, 2017a,b). Briefly, mice were anesthetized with isoflurane (5%, 1 l/min flow rate) and placed in a stereotaxic frame (Narishige). A Quintessential Stereotaxic Injector (Stoelting) was used to inject virus particles at a speed of 50–100 nl/min. The pipette (Drummond) was slowly withdrawn 5 min after injection. Fourteen to twenty-five days after viral injections, animals were used for (i) mapping local and long-range projections, (ii) slice electrophysiology, (iii) $Ca^{2+}$ imaging, and (iv) behavioral tests. AAV8-EF1a-double floxed-hChR2(H134R)-mCherry-WPRE-HGHpA (Addgene Viral Service, #20297) was unilaterally injected on LC (100 nl; coordinates from bregma: AP: −5.4 mm, DV: −3.7 mm, ML: ±0.9 mm) of *Scgn*-Cre mice. For behavioral testing, AAV8-hSyn-DIO-hM3D (Gq)-mCherry (Addgene, #44361) or AAV8-hSyn-DIO-hM4D(Gi)-mCherry (Addgene, #44362) was bilaterally injected in LC (125 nl/side; coordinates relative to bregma: AP: −5.4 mm, DV: −3.7 mm, ML: ±0.9 mm) of *Scgn*-Cre mice. Likewise, AAV8-hSyn-DIO-hM3D (Gq)-mCherry or pAAV-hSyn-DIO-mCherry (Addgene, #50459) was unilaterally injected into the PVN (40 and 25 nl, respectively, coordinates relative to bregma: AP: −0.7 mm, DV: −4.8 mm, ML: ±0.2 mm) for *ex vivo* electrophysiology and projection mapping. For $Ca^{2+}$ imaging, LC neurons projecting to the mPFC were retrogradely loaded by using AAV-hSyn1-GCaMP6f-P2A-nls-dTomato (Addgene, #51085, 300 nl, coordinates relative to bregma: AP: +1.8 mm, DV: −2.6 mm, ML: ±0.25 mm).

### Anterograde tracing

Rats ($n = 4$ at 12 weeks of age) received an injection of biotinylated dextran amine (BDA, 10,000 Da, 10%, 0.2 μl injection volume; Molecular Probes) into the locus coeruleus (LC) under deep anesthesia (coordinates: 1.5 mm lateral, +9.7 mm caudal to bregma, and −6.0 mm ventral from the dural surface) with their heads fixed in a stereotaxic frame. Seven days after surgery, the animals were transcardially perfused and their brains processed for immunohistochemistry.

### Retrograde tracing

Rats ($n = 4$ at 12 weeks of age) were deeply anesthetized and their head fixed in a stereotaxic frame and biotinylated dextran amine (3,000 MW, 2 μl, 10%; Molecular Probes) infused in the PFC unilaterally at the coordinates: AP: 0.5 mm, DV: 2.5 mm, and ML: −4.5 mm. After a 7-day survival period, animals were transcardially perfused and their brains processed for immunohistochemistry.

### Ventricular administration of CNTF via icv cannula

For the experiments, rats ($n = 14$ at 12 weeks of age) and transgenic mice [$Cntf^{-/-}$ ($n = 10$), $Scgn^{-/-}$ ($n = 10$), and corresponding littermates ($n = 10$ each)] were used. To administer CNTF in awake animals, an icv cannula (PlasticsOne) was introduced into the lateral ventricle and fixed to the skull by using $Zn_3(PO_4)_2$ cement (Adhesor; Pentron). A guide cannula was placed into the brains of both rats

(4 mm long, i.d. × o.d. = 0.39 × 0.71 mm, at AP: −0.9 mm, ML: +1.4 mm, DV: −4.0 mm) and mice (3 mm long, i.d. × o.d. = 0.39 × 0.71 mm, t AP: −0.2 mm, ML: +1.0 mm, DV: −3.0 mm) under deep anesthesia. After 7 days, rats received CNTF ($n = 7$, 4 µl, 6 ng/µl) or artificial cerebrospinal fluid ($n = 7$, aCSF, 4 µl) through an internal cannula (4.5 mm long, i.d. × o.d. = 0.18 × 0.36 mm). In mice ($n = 5$/genotype for CNTF or aCSF), drug infusion (3 µl) was made by using internal cannulas with specifications as follows: 3 mm long, i.d. × o.d. = 0.39 × 0.71 mm;  and  3.5 mm long, i.d. × o.d. = 0.39 × 0.71 mm. Sixty minutes after CNTF or vehicle infusion, animals were subjected to an open-field test. Subsequently, animals were decapitated and their medullopontine brainstem and mPFC removed and processed for protein (Western blotting) and catecholamine (HPLC) analysis, respectively.

### Sampling of cerebrospinal fluid (CSF)

Formalin-stressed ($n = 4$) and control ($n = 4$) rats (12 weeks of age) were deeply anesthetized and their head fixed into a stereotaxic frame with maximal anteflexion. To approach the 4th ventricle, the skin was incised, nuchal muscles pulled to the sides, and the bony tissue below the cerebellum removed using a surgical drill. The lamina epithelialis was identified as being immediately caudal to the cerebellum and pierced with a 26-G syringe. Subsequently, 15 µl CSF was removed from the 4th ventricle using a standard 20-µl laboratory pipette (Eppendorf; Cottrell et al, 2004).

### Transcardial perfusion

Animals were routinely perfusion-fixed with 4% PFA in 0.1M PB (pH 7.4) under deep anesthesia as described (Alpár et al, 2014; Lendvai et al, 2013; Mulder et al, 2009; Romanov et al, 2015).

### Immunohistochemistry, microscopy, and imaging

Free-floating sections (30 µm) were rinsed in PB (pH 7.4) and pre-treated with 0.3% Triton X-100 (in PB) at 22–24°C for 1 h to enhance the penetration of antibodies (for references, see Alpár et al, 2004, 2014; Lendvai et al, 2013; Mulder et al, 2009; Ong et al, 2014; Romanov et al, 2015; Severi et al, 2012; Severi et al, 2013). Non-specific immunoreactivity was suppressed by incubating our specimens in a cocktail of 5% normal donkey serum (NDS; Jackson ImmunoResearch), 2% bovine serum albumin (BSA; Sigma), and 0.3% Triton X-100 (Sigma) in PB at 22–24°C for 1 h. Sections were then exposed (16–72 h at 4°C) to select combinations of primary antibodies (Tools and Reagents Table) diluted in PB to which 0.1% NDS and 0.3% Triton X-100 had been added. We used tyramide signal amplification (PerkinElmer) to detect CNTF receptor immunoreactivity. After extensive rinsing in PB, sections were processed by using chromogenic or immunofluorescence detection methods as described previously (Lendvai et al, 2013). In single labeling experiments using chromogenic amplification, biotinylated anti-rabbit IgG raised in donkey (1:1,000; Jackson ImmunoResearch, at 22–24°C for 2 h) was used as secondary antibody followed by exposure to pre-formed avidin–biotin complexes also incorporating horseradish peroxidase at 22–24°C for 1 h. Immunosignals were visualized by 3,3′-diaminobenzidine (Sigma, 0.025%) as chromogen intensified with Ni-ammonium sulfate (Merck, 0.05%) in the presence of 0.001% $H_2O_2$ as substrate (dissolved in 0.05 M Tris buffer, pH 8.0). In multiple immunofluorescence labeling experiments, immunoreactivities were revealed by carbocyanine (Cy) 2-, 3-, or

5-tagged secondary antibodies raised in donkey [1:200 (Jackson ImmunoResearch), at 22–24°C for 2 h]. In human samples, lipofuscin autofluorescence was quenched by applying Sudan Black B (1%, dissolved in 70% ethanol; Schnell et al, 1999). Glass-mounted sections were coverslipped with glycerol/gelatin (GG-1; Sigma). Results of chromogenic histochemistry were captured on a NIKON Eclipse Microscope. The sections processed for multiple immunofluorescence histochemistry were inspected and images acquired on a LSM780 confocal laser-scanning microscope (Zeiss) with optical zoom ranging from 1× to 3× at 63× primary magnification (Plan-Apochromat 63×/1.40), and pinhole settings limiting signal detection to 0.5–0.7 µm "optical thickness". Emission spectra for each dye were limited as follows: Cy2 (505–530 nm), Cy3 (560–610 nm), and Cy5 (650–720 nm). VGLUT2[+] and GRIA1[+] profiles were 3D-reconstructed using the cell surface reconstruction tool of Imaris x64 (version 9.0.2, BitPlane). Multi-panel figures were assembled in CorelDraw X7 (Corel Corp.).

### Electron microscopy

Animals were perfusion-fixed with 4% PFA and 0.1% glutaraldehyde in 0.1 M PB and their brains dissected out and cut on a vibratome (Leica V1200S). Sections were osmificated, dehydrated, and embedded in Durcupan (Fluka, ACM). Sixty-nanometer-thick ultrathin sections were prepared on an Ultracut UCT ultramicrotome (Leica) and analyzed on a Tecnai 10 electron microscope.

### Quantitative image analysis

Images from serial sections ($n = 4$ per animal and condition) were captured using a Zeiss LSM880 confocal laser-scanning microscope. CRH[+] boutons or GFAP[+]/ZsGreen1[+] cells were identified offline by an experimenter blinded to the case conditions and their density calculated in demarcated regions of interest (ROIs) using the ZEN software (Zeiss). In all cases, histochemical specimens were also pre-coded to minimize bias. ROIs were defined as a 15-µm-wide periventricular rim around the 3rd ventricle. Thereafter, the length of the ventricular surface was measured (in µm) with GFAP[+]/ZsGreen1[+] cell numbers or CRH[+] bouton numbers normalized and expressed per mm or per 100 µm², respectively. CNTFR[+] boutons were counted along the perimeter of BDA[+] (retrogradely labeled)/TH[+] and BDA[−]/TH[+] LC neurons. The perimeter of neurons was measured using ZEN2010 (Zeiss), expressed as µm, and used to normalize the density of CNTFR[+] profiles.

### Immunoprecipitation

Brainstems were isolated from 12-week-old rats ($n = 3$) and collected in lysis buffer containing 50 mM NaCl, 20 mM HEPES, 10 µM CaCl₂, 0.2% Triton X-100, and a cocktail of protease inhibitors (Roche; pH was adjusted to 7.4). Tissues were homogenized by sonication and centrifuged at 18,000 $g$ for 30 min. Only supernatants were used in subsequent experiments. Samples (50 µl) were incubated with rabbit anti-secretagogin primary antibody (1:2,000; provided by L. Wagner) overnight at 4°C. An aliquot of each sample was probed in parallel with rabbit IgG (2 µg/50 µl; Santa Cruz) to control for non-specific binding. Subsequently, samples were incubated with protein G Dynabeads (Novex; Life Technologies) for 90 min. After repeated rinses, Dynabeads were collected and bound proteins eluted with Laemmli buffer and separated on 10% resolving gels under denaturing conditions (SDS–PAGE). LC micropunches from human subjects deceased

due to acute heart failure were processed according to a protocol identical to that used for rat samples.

### Western blotting

Protein samples were prepared from both rat ($n = 8$) and mouse ($n = 12$) brain stems by natively cutting them at 300 μm thickness on a Leica 1850 cryostat. Consecutive sections (six in rats, four in mice) containing the locus coeruleus (LC) were collected to form a sample. Medial prefrontal cortices were excised on a pre-chilled surface under a stereomicroscope. Tissues were sonicated in TNE buffer containing 0.5% Triton X-100 (Sigma), 1% octyl-β-D-gluco-pyranoside (Calbiochem), 5 mM NaF, 100 μM $Na_3VO_4$, and a mixture of protease inhibitors (Complete™; Roche). Cellular debris and nuclei were pelleted by centrifugation (800 $g$, 4°C, 10 min). Protein concentrations were determined by Bradford's colorimetric method (Bradford, 1976). Samples were diluted to a final protein concentration of 2 μg/μl, denatured in 5× Laemmli buffer, and analyzed by SDS–PAGE on 8 or 10% resolving gels. After transfer onto Immobilon-FL PVDF membranes (Millipore), membrane-bound proteins were blocked (1.5 h) with 3% BSA and 0.5% Tween-20 in Tris-buffered saline and subsequently exposed to primary antibodies (Tools and Reagents Table) at 4°C overnight. Combinations of HRP-conjugated secondary antibodies (from goat, rabbit, or mouse hosts, Jackson ImmunoResearch, 1:10,000, 2 h) were used for signal amplification. Image acquisition and analysis were performed on a Bio-Rad XRS$^+$ imaging platform equipped with Image Lab 3.01 software (Bio-Rad Laboratories). β-Actin (1:10,000; Sigma) was used as loading control.

### Synaptosomal proteins

Rats ($n = 2$, 12 weeks of age) were perfused with ice-cold saline with their brains rapidly dissected out. The medial prefrontal cortex was excised and homogenized in 0.32 M sucrose-containing HEPES buffer (in mM: 145 NaCl, 5 KCl, 2 $CaCl_2$, 1 $MgCl_2$, 5 glucose, and 5 HEPES at pH 7.4). After centrifugation, the supernatant was repeatedly cleared (by centrifugation at 15,000 $g$ in 1.3 M sucrose-containing HEPES buffer). The resultant pellet, enriched in synaptic proteins (Ramos-Ortolaza et al, 2010), was used to probe the presynaptic enrichment of secretagogin.

### Quantitative PCR

RNA was extracted using the RNeasy mini kit (Qiagen) with a DNase I step performed to eliminate traces of genomic DNA, and reverse-transcribed using a high-capacity cDNA reverse transcription kit (Applied Biosystems). Reactions were performed after an initial denaturation step at 95°C for 2 min followed by 40 cycles of 95°C for 1 min denaturation, annealing and extension at 60°C (1 min each), and a dissociation stage at 72°C (2 min) on a CFX 96 apparatus (Bio-Rad). Primer pairs amplified short fragments of the *Cntf* gene (*forward*: 5′-ATGGCTTTCGCAGAGCAAAC-3′, *reverse*: 5′-CAACGAT CAGTGCTTGCCAC-3′). Samples without reverse transcriptase served as negative controls. In select experiments, amplicons were resolved on 1% agarose gels (Appendix Fig S1C).

### RNAi-mediated gene silencing in vitro

Knock-down of *Scgn* mRNA expression in INS-1E cells was through the application of *Scgn* siRNA (Santa Cruz, 250 pmol/500 μl, diluted and transfection carried out using JETPRIME reagents) for 48 h. To induce tyrosine hydroxylase phosphorylation, recombinant CNTF was added (40 ng/μl; Sigma) for 48 h. INS-1E cells were lysed and processed for Western blotting.

### RNAi-mediated gene silencing in vivo

Male rats (12 weeks of age) received icv injections of either SMART-pool Accell *Cntf* siRNA ($n = 8$; 4 μl volume, 1 nmol siRNA in total) or non-targeting siRNA ($n = 8$) as described previously (Romanov et al, 2015). After 4 days, rats ($n = 4$) were subjected to formalin stress (4% PFA, 50 μl injection volume) or were left undisturbed in their home cages ($n = 4$). After 60 min of PFA administration, animals were decapitated and their brains removed with the medullary–pontine brainstem and mPFC isolated and subjected to protein analysis by Western blotting. The remaining mPFC samples were subjected to HPLC analysis.

In a parallel series of experiments, the LC of male mice (12 weeks of age) was targeted stereotaxically by either SMARTpool Accell *Scgn* siRNA ($n = 3$; 4 μl volume, 1 nmol siRNA in total) or non-targeting siRNA ($n = 3$). After 3 days, mice were subjected to formalin stress, by decapitation after 60 min. The medullary–pontine brainstem and bilateral mPFCs were used for Western blotting.

### High-performance liquid chromatography

$Scgn^{-/-}$ and wild-type littermates ($n = 12$ in total, all 10–12 weeks old) were sacrificed and their brains extracted and frozen on dry ice within 30 s. The medial prefrontal and parietal cortices (the latter equivalent to the S1 somatosensory area) were dissected from frozen coronal slices on a cold plate (at constant −10°C). Samples were ultrasonicated in 25 volumes of perchloric acid, Na-bisulfite, and 3,4-dihydroxybenzylamine as internal standard (final concentration 0.1 M, 0.4 mM, and 5 μg/l, respectively) and centrifuged at 16,100 $g$ for 10 min. For the determination of tissue levels of norepinephrine (NE), dopamine (DA), and 3,4-dihydroxyphenylacetic acid (DOPAC), supernatants were extracted with aluminum oxide and injected into a HPLC system with electrochemical detection as described (Pifl et al, 1991) with minor modifications (LiChroCART® 250-4, RP18μ, 5-μm column, HP Programmable Electrochemical Detector 1049A, and a mobile phase with 9% methanol).

### Single-cell RNA-seq

C57Bl6/N juvenile mice (P14–28) of both sexes in control were used for single-cell collection as described (Romanov et al, 2017a,b). The processing of cells from male and female animals was random to minimize methodological bias. Mice were deeply anesthetized (5% isoflurane) and transcardially perfused with 40 ml ice-cold pre-oxygenated (95% $O_2$/5% $CO_2$) cutting solution containing (in mM) 90 NaCl, 26 $NaHCO_3$, 2.5 KCl, 1.2 $NaH_2PO_4$, 10 HEPES-NaOH, 5 Na-ascorbate, 5 Na-pyruvate, 0.5 $CaCl_2$, 8 $MgSO_4$, and 20 glucose. A central column of the mouse hypothalamus spanning the posterior preoptic area to the arcuate nucleus along its rostrocaudal axis, paraventricular nucleus (dorsally), and the ventrolateral hypothalamic area (laterally) was microdissected from serial 300-μm-thick coronal slices under microscopy guidance and then dissociated using the Papain Dissociation System (Worthington). Thus, isolated single cells included those that line the anterior extent of the 3$^{rd}$ ventricle. Cells were concentrated by centrifugation to a density of 600–1,000 cells/μl. After mixing C1 suspension reagent (4 μl; Fluidigm) with the cell suspension (7 μl), this mixture was loaded into

a C1-AutoPrep IFC microfluidic chip designed for cells 10–17 μm in diameter (Fluidigm) and processed on a Fluidigm C1 instrument using the mRNA Seq: Cell Load (1,772×/1,773×) script (30 min at 4°C). The microfluidic plate was then transferred to an automated microscope (Nikon TE2000E) to acquire a bright-field image of each capture site at 20× magnification using μManager (https://micro-manager.org/) in < 15 min. Quality control for exclusion of debris or doublets was performed after each capture experiment. Following lysis, cDNA synthesis, amplification, and tagmentation, high-throughput RNA sequencing was performed on an Illumina HiSeq 2000 sequencer (Islam *et al*, 2014). Next, the dataset was processed with the BackSpinV2 algorithm (Romanov *et al*, 2017a,b) and first grouped for main cell lineages. Ependymal cells were separated using genes associated with motile cilia, such as the *Enkur* and *Foxj1* gene pair (Romanov *et al*, 2017a,b; Roy, 2009; Zeisel *et al*, 2015). Here, we focused on neurotrophins, ionotropic and metabotropic receptors for glutamate and GABA, and neuropeptide receptors given that "stress-on" CRH neurons in the paraventricular nucleus always contain a fast neurotransmitter in conjunction with CRH (Romanov *et al*, 2017a,b). Data were then rendered as heatmaps (GenePattern) for improved visual clarity.

## Neurological assessment

Behavioral tests were conducted by the same experimenter—who was blinded to mouse phenotypes—in an isolated room and at the same time of day throughout. Three- to six-week-old mice [$n = 9$ (wild-type) and $n = 13$ $Scgn^{-/-}$] were used. The animals were housed in groups of 3–5 per cage in standard macrolon cages under a 12-h/12-h light/dark schedule (lights on at 08:00) in controlled environmental conditions of humidity (50–65%) and temperature ($22 \pm 2°C$) with food and water available *ad libitum*. All experimental procedures were approved by the local ethical committee of the Medical University of Vienna (BMWFW-66.009/0277-WF/V/3b/2017) and also met European legislative requirements. A modified SHIRPA protocol (Rogers *et al*, 2001) was used as comprehensive phenotyping tool. Herein, data on provoked sensory performance were presented as measures of stress responsiveness. First, mice were transferred to an arena (56 × 34 cm) for observation of sensory functions, including their pinna reflex response/refraction score (while the mouse being gently restrained, each auditory meatus was lightly touched with either ear retraction or head movement recorded at a scale of "0" = no response, "1" = active retraction, moderately brisk flick, "2" = hyperactive, repetitive flick). Foot withdrawal was assayed after lifting the mice by their tail and applying gentle lateral compression of the middle digit of their hind paw using a fine forceps (scoring: "0" = no response, "1" = slight withdrawal, "2" = moderate withdrawal, "3" = rapid withdrawal, "4" = rapid, brisk withdrawal with extension/flexion). Provoked biting was scored by approaching each gently immobilized mouse from the frontal direction with a cotton tab ("0" = present, "1" = absent, "2" = active exploration and embracing instead of defense). Subsequently, behavioral scores were ranked and processed statistically. Another cohort of mice ($n = 6/6$ $Scgn^{-/-}$/wild-type littermates, all 12–16 weeks old) were placed into a programmable foot-shock chamber to habituate for 10 min each. The subsequent day, $n = 3$ from both genotypes received a series of five foot-shocks (0.8 mA) each during a period of 10.5 min with random intervals ranging from 90 to 120 s. Control animals ($n = 3$

each) spent equivalent periods in the chamber without any manipulation. We used EthoVision (Noldus) to analyze the following parameters: (i) average latency to first movement after foot-shock (only for mice from shock-exposed groups; data represent average latency/mouse), (ii) total distance covered, and (iii) time spent at center ("cumulative duration") during 10.5 min.

The von Frey test, a mechanical non-invasive nociceptive assay, was used to evaluate sensory threshold in $Scgn^{-/-}$ and in wild-type littermates. Animals were placed on a metal frame, and their left hind paw was pressed from below with a series of Nylon rods (Touch Test Sensory Evaluator, EXACTA, North Coast Medical) corresponding to increasing applied weight. The sensory threshold of the animal was identified as the first rod which evoked withdrawal of the extremity.

## Open-field and elevated plus maze tests

To determine the acute effect of icv-infused CNTF on open-field behaviors, animals were transferred to a rectangular arena of either 90 × 90 cm for rats or 40 × 40 cm for mice 60 min after CNTF application through a guide cannula. Spontaneous locomotor behaviors were recorded with an overhead camera for 10 min. Video recordings were analyzed offline using the "Animal Tracker" plug-in developed for ImageJ (Gulyas *et al* 2016), including a virtual "spit arena" made up of central and peripheral areas, the latter defined as 8 and 18 cm from the wall for mice and rats, respectively. The time spent in each area, as well as the distance covered, was measured.

For *Scgn*-Cre mice, a rectangular open-field arena of 70 × 70 cm footprint with 30-cm walls was used with assays lasting for 5 min. Subsequently, anxiety-like behavioral phenotypes were determined in an elevated plus maze with uniform dimensions for its open and closed arms (32 cm in length and 5 cm in width) for 3 min. The apparatus was elevated 76.5 cm above the floor surface. A Smart Video Tracking System (Harvard Apparatus) was used for data processing offline. CNO was administered i.p. at a dose of 2 mg/kg of body weight to activate DREADD constructs *in vivo*, 15–20 min before each test. In both tests, animals were handled by an experimenter blinded to the case condition to ensure objectivity.

## Ex vivo *electrophysiology and Ca²⁺ imaging*

We used 3- to 4-week-old mice ($n = 4$–5/genotype) and a protective recovery method for acute brain slice preparation (Zhao *et al*, 2011). Briefly, mice were deeply anesthetized and perfused with 20 ml ice-cold pre-oxygenated (95% $O_2$/5% $CO_2$) solution containing (in mM) 93 N-methyl-D-glutamine–HCl, 30 $NaHCO_3$, 2.5 KCl, 1.2 $NaH_2PO_4$, 20 HEPES-NaOH, 5 Na-ascorbate, 3 Na-pyruvate, 0.5 $CaCl_2$, 8 $MgSO_4$, and 25 glucose (pH 7.4). Brains were rapidly extracted and immersed in the same solution. Subsequently, 300-μm-thick coronal slices of the prefrontal cortex were cut on Leica VT1200S vibratome. Slices were then transferred to a recovery chamber filled with the same solution (32°C) for 12 min and later kept (minimum 60 min prior to the recordings) in a solution containing (in mM) 90 NaCl, 26 $NaHCO_3$, 3 KCl, 1.2 $NaH_2PO_4$, 20 HEPES-NaOH, 5 Na-ascorbate, 3 Na-pyruvate, 1.5 $CaCl_2$, 2 $MgSO_4$, 0.5 L-glutathione, and 25 glucose (pH 7.4).

Patch-clamp recordings as well as $Ca^{2+}$ imaging of ependymal cells lining the 3$^{rd}$ ventricle, NE neurons in the LC, and layer 5 pyramidal neurons in the mPFC (at ~+1.3 to +2.2 mm relative to bregma and prelimbic and infralimbic cortices) were performed in

oxygenated (95% $O_2$/5% $CO_2$) artificial CSF containing (in mM) 124 NaCl, 2.5 KCl, 2 $MgCl_2$, 1.5 $CaCl_2$, 24 $NaHCO_3$, 1.2 $NaH_2PO_4$, 5 HEPES, and 12.5 glucose. Patch pipettes with a resistance of 3–5 MΩ contained (in mM) 120 K-gluconate, 6 KCl, 10 HEPES-KOH, 5 EGTA, 4 ATP-Mg, and 0.3 GTP (pH was adjusted to 7.3 with KOH). Perfusion speed was set to 2.5–3 ml/min. To record base current and spontaneous excitatory postsynaptic currents (sEPCs) in control and after application of 10 μM L-norepinephrine (Sigma), pyramidal cells were clamped at −70 mV. CNO was used at a concentration of 10 μM to activate DREADDs *ex vivo*. Data were analyzed using Clampfit 10.0 (Molecular Devices), MiniAnalysis (Synaptosoft), and SigmaPlot (Systat).

$Ca^{2+}$ imaging on retrogradely labeled NE neurons was done on an AxioExaminer.D1 microscope (Zeiss) equipped with a water-immersion 20× Plan-APOCHROMAT objective (Zeiss) and a Cool-SNAP HQ$^2$ camera (Photometrics). Illumination was provided by a VisiChrome monochromator (Visitron Systems) to excite GCaMP6f with data recorded and processed in the VisiView software (Romanov *et al*, 2017b).

*Ex vivo* optogenetics employed excitation switches between 470 nm and 535 nm on a pE-100 CoolLED illumination system (CoolLED) at a uniform light intensity of 0.2 mW measured at the tissue surface.

### Statistical analysis

Data were analyzed using the Statistical Package for the Social Sciences (version 21.0, SPSS Inc.). Integrated optical densities of immunoreactive targets in Western blotting or immunohistochemical experiments were evaluated using Student's *t*-test (on independent samples). Drug effects in electrophysiology studies were defined by either one-way ANOVA or paired Student's *t*-test as appropriate. The behavior of knock-out animals in the open-field and elevated plus maze tests was evaluated by ANOVA using a general linear model with genotype, sex, and treatment being fixed factors followed by appropriate group comparisons. Behavioral data on pairs of treatment groups were routinely analyzed using the Mann–Whitney *U*-test (one-tailed where deemed appropriate) or Student's *t*-test. Data are expressed as means ± s.e.m. A *P*-value of < 0.05 was considered statistically significant.

## Data availability

Raw data files for single-cell RNA sequencing have been deposited in NCBI Gene Expression Omnibus under accession number GEO: GSE74672 (https://www.ncbi.nlm.nih.gov/geo/query/acc.cgi?acc = GSE74672).

**Expanded View** for this article is available online.

## Acknowledgements

We thank A. Reinthaler, E. Borok, P. Rebernik, and A. Németh for their expert laboratory assistance; F. Girach for her initial help with *cfos*-Cre$^{ERT2}$::*ROSA26-stop-ZsGreen1^{f/f}* mice; H. Reither, A. Wolf, and E. Dögl for HPLC analysis; and B. Törőcsik and A. Dobolyi for constructive discussions. A. Zeisel and S. Linnarsson are acknowledged for their initial processing of single-cell RNA-seq data. Anti-secretagogin antibody, INS-1E cells, and *Crh-Ires*-Cre mice were kindly provided by L. Wagner, P. Maechler, and J.S. Bains, respectively. This work was supported by the National Brain Research Program of Hungary (KTIA_NAP_13-2014-0013,

A.A.; KTIA_NAP_13-1-2013-0001, M.P.; and 2017-1.2.1-NKP-2017-00002, A.A.); Excellence Program for Higher Education of Hungary (FIKP-2018, A.A.); Swedish Research Council (T.G.M.H., T.H.); Novo Nordisk Foundation (T.G.M.H., T.H.); Hjärnfonden (T.H.); European Research Council (SECRET-CELLS, 2015-AdG-695136; T.H.); intramural funds of the Medical University of Vienna (T.H.); and NIH grant AG051459 (T.L.H.).

## Author contributions

TH and AA conceived the project; AA, TGMH, TLH, and TH procured funding; AA, PZ, TGMH, RAR, TLH, and TH designed experiments; PZ, JH, and ZH performed quantitative neuroanatomy and circuit tracing; CP determined monoamine levels; RAR, SK, and MB conducted and analyzed patch-clamp electrophysiology and chemogenetics; JP, IS, and AG contributed *Cntf*$^{-/-}$ mice and performed neuroanatomy and behavioral assessment; JB and AA contributed to circuit tracing; PL, AGM, and RAR demonstrated ependymal cell activation using TRAP and histochemistry; ET and RAR conducted and analyzed single-cell RNA-seq data; GA and GZ assayed behavioral phenotypes; EK, ROT, GSp, KM, ZM, FE, GSz, and GL contributed unique reagents and transgenic mouse lines; MP contributed human subject cohort; and AA and TH wrote the manuscript with input from all co-authors.

## Conflict of interest

The authors declare that they have no conflict of interest.

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
