## [Review Process File · The EMBO Journal]

Hypothalamic CNTF volume transmission shapes cortical noradrenergic excitability upon acute stress

Alán Alpár, Péter Zahola, János Hanics, Zsófia Hevesi, Solomiia Korchynska, Marco Benevento, Christian Pifl, Gergely Zachar, Jessica Perugini, Ilenia Severi, Patrick Leitgeb, Joanne Bakker, Andras G. Miklosi, Evgenii Tretiakov, Erik Keimpema, Gloria Arque, Ramon O. Tasan, Günther Sperk, Katarzyna Malenczyk, Zoltán Máté, Ferenc Erdélyi, Gábor Szabó, Gert Lubec, Miklós Palkovits, Antonio Giordano, Tomas G.M. Hökfelt, Roman A. Romanov, Tamas L. Horvath, Tibor Harkany

Review timeline:

Submission date:	21st Jun 2018
Editorial Decision:	24th Jul 2018
Revision received:	5th Aug 2018
Accepted:	13th Aug 2018

Editor: Karin Dumstrei

Transaction Report:

1st Editorial Decision

24th Jul 2018

Thank you for transferring your manuscript with referee reports from another journal to The EMBO Journal. A good expert in the field has now reviewed your study, the referee reports from the previous journal and the point-by-point response.

As you can see from the comments below, the referee finds the study interesting and suitable for publication in The EMBO Journal. There are just a few minor comments to resolve before acceptance here - please see specific referee comments below.

When you resubmit the revised version would also take care of the following items:

- Please take a look at our author guidelines regarding supplemental figures
<http://emboj.embopress.org/authorguide#expandedview>

- We now encourage the publication of source data, particularly for electrophoretic gels and blots, with the aim of making primary data more accessible and transparent to the reader. It would be great if you could provide me with a PDF file per figure that contains the original, uncropped and unprocessed scans of all or key gels used in the figure? The PDF files should be labeled with the appropriate figure/panel number, and should have molecular weight markers; further annotation could be useful but is not essential. The PDF files will be published online with the article as supplementary "Source Data" files.

REFeree COMMENTS

Referee #1:

The study by Alpár and colleagues analyzes the link between the stress-induced hypothalamus-pituitary-adrenal axis (HPA) activity and the increased cortical alertness, providing a new and comprehensive picture about the molecular chain of events connecting the two processes. In particular, the authors found that corticotropin-releasing hormone (CRH)-containing (+) neurons of the hypothalamus can stimulate (through glutamate release) ependymal cells of the third ventricle to release ciliary neurotrophic factor (CNTF) into the brain's aqueductal system. This release leads to a long-range activation of noradrenergic neurons (NA) located in the locus coeruleus (LA) innervating the prefrontal cortex, via a sequential phosphorylation including the Ca²⁺-sensor secretagoin. The work is excellent, very well written and relies on a combination of cutting-edge in vivo technology based on opto-/chemogenetic, electrophysiology, biochemistry and imaging approaches. Every single statement of the work is supported by compelling evidences provided by a precise manipulation of the biological system analyzed. Moreover, the authors provide evidence that the connection between HPA activity, noradrenergic neurons of LA and behavioral moderations is a molecular mechanism conserved amongst the evolution. For all these reasons, the present work deserves publication in EMBO journals.

The authors should address the following minor points:

- 1) In figure 1D, the authors perform electrophysiological recordings on ependymal cells to demonstrate the presence of glutamate-dependent synaptic events. It would be interesting to report the electrophysiological parameters of these events (frequency, amplitude, decay and rise times) in the supplementary figure 1.
- 2) I would suggest to remove the "personal communication" stating that ependymal cells is the main source of CNTF in the brain. Indeed, the single-cell RNA-seq analyses showed in figure 1C provides a sufficient demonstration that these cells express CNTF. Evidences reported in literature showed CNTF to be expressed in other brain cells, including white matter astrocytes of the optical tract (Dallner et al., GLIA 2002).

1st Revision - authors' response

24th Jul 18

Thank you for your positive and constructive comments on our submission. We were particularly glad to learn that you have found our manuscript "comprehensive, excellent, very well written and relies on a combination of cutting-edge in vivo technology".

In accord with your specific (and more minor) queries, we have revised the manuscript as follows.

Q1: "In figure 1D, the authors perform electrophysiological recordings on ependymal cells to demonstrate the presence of glutamate-dependent synaptic events. It would be interesting to report the electrophysiological parameters of these events (frequency, amplitude, decay and rise times) in the supplementary figure 1."

We appreciate your attention to detail. These data were added to "Expanded View Figure 1" as its panel "D".

Q2: "I would suggest to remove the "personal communication" stating that ependymal cells is the Main source of CNTF in the brain. Indeed, the single-cell RNA-seq analyses showed in figure 1C provides a sufficient demonstration that these cells express CNTF. Evidences reported in literature showed CNTF to be expressed in other brain cells, including white matter astrocytes of the optical tract (Dallner et al., GLIA 2002)."

We perhaps were unclear in phrasing this statement since it intended to specifically refer to ependymal/stem cell and neuronal sources of CNTF. We certainly do not contest glial origins for this trophic factor. Therefore, and to satisfy your query, we have removed the erroneous statement.

Corresponding Author Name: Tibor HARKANY

Journal Submitted to: The EMBO Journal

Manuscript Number: EMBOJ-2018-100087R1